# Boosting urea electrooxidation on oxyanion-engineered nickel sites via inhibited water oxidation

Xintong Gao[1,2], Xiaowan Bai [1,2], Pengtang Wang[1,2], Yan Jiao [1], Kenneth Davey [1], Yao Zheng [1]✉ & Shi-Zhang Qiao [1]✉

Renewable energy-based electrocatalytic oxidation of organic nucleophiles (e.g.methanol, urea, and amine) are more thermodynamically favourable and, economically attractive to replace conventional pure water electrooxidation in electrolyser to produce hydrogen. However, it is challenging due to the competitive oxygen evolution reaction under a high current density (e.g., >300 mA cm$^{-2}$), which reduces the anode electrocatalyst's activity and stability. Herein, taking lower energy cost urea electrooxidation reaction as the model reaction, we developed oxyanion-engineered Nickel catalysts to inhibit competing oxygen evolution reaction during urea oxidation reaction, achieving an ultrahigh 323.4 mA cm$^{-2}$ current density at 1.65 V with 99.3 ± 0.4% selectivity of N-products. In situ spectra studies reveal that such in situ generated oxyanions not only inhibit OH$^-$ adsorption and guarantee high coverage of urea reactant on active sites to avoid oxygen evolution reaction, but also accelerate urea's C − N bond cleavage to form CNO$^-$ intermediates for facilitating urea oxidation reaction. Accordingly, a comprehensive mechanism for competitive adsorption behaviour between OH$^-$ and urea to boost urea electrooxidation and dynamic change of Ni active sites during urea oxidation reaction was proposed. This work presents a feasible route for high-efficiency urea electrooxidation reaction and even various electrooxidation reactions in practical applications.

Abundant urea in wastewater and human urine is a practically promising alternative energy carrier and (indirect) hydrogen (H$_2$) storage chemical[1–4]. The six-electron transferred electrocatalytic urea oxidation reaction (UOR) is an important anode half-reaction in energy-related applications including, direct urea /urine fuel cells (DUFC)[5–7] and urea-assistant water electrolysers[8–10]. The thermodynamic potential for UOR coupled with cathode hydrogen evolution reaction is theoretically 0.37 V (vs. reversible hydrogen electrode, RHE), a value significantly less than that of 1.23 V for water electrolysis[8,10]. Therefore, UOR has a significant potential to reduce voltage input for H$_2$ production by replacing traditional oxygen evolution reaction (OER) in the anode of a water electrolyser. Recently, nickel (Ni)-based electrocatalysts have been reported for UOR[11–15]. In most cases, these catalysts undergo surface reconstruction into nickel oxyhydroxide (NiOOH) species during UOR because of the anodic potential and electrolyte's hydroxyl anions, which serve as the active species for UOR[11,14,16]. However, the derived NiOOH are also highly active species for OER, an undesired side reaction toward UOR[17,18]. More importantly, OER will over-oxidize NiOOH to high-valent nickel[19,20], thereby degrading the electrode stability and reducing further UOR efficiency. This competition between UOR and OER is more significant in practical UOR-related applications, especially with high operating potential and large current density devices[3,14,21].

---

[1]School of Chemical Engineering, The University of Adelaide, Adelaide, SA, Australia. [2]These authors contributed equally: Xintong Gao, Xiaowan Bai, Pengtang Wang. ✉e-mail: yao.zheng01@adelaide.edu.au; s.qiao@adelaide.edu.au

The competition between UOR and OER originates from the adsorption of OH⁻. UOR in an alkaline environment needs OH⁻ to proceed through a series of proton-coupled electron transfers and to cleavage the C−N bonds in urea molecule[21,22], however, excessive OH⁻ adsorption on the active sites will block adsorption of urea reactant and accelerate competing OER. Although this kind of competition has been mentioned in some anodic reactions including, methanol electrooxidation and 5-hydroxymethylfurfural electrooxidation[23,24], it has not been explored in UOR. Importantly, there is no efficient strategy for catalysts to overcome this competition. Practically, given UOR and OER is six-electron and four-electron transferred reaction, respectively[3,16,25], competitive OER will, theoretically, reduce anode current density by >30%. Therefore, the design for catalysts to obviate OER in the UOR at a high potential to achieve a large apparent current density and excellent electrode stability is essential for the development of the next generation of urea-assisted industrial water electrolysers.

Here, we realized highly selective and active UOR on oxyanions (sulfur, phosphorus, and selenium) coordinated nickel catalyst. Electrochemical kinetics measurements, combined with in situ spectroscopy characterizations and potential dependent density function theory (DFT) calculations indicate that the critical roles of such in situ generated oxyanions: (1) it protects Ni active sites with high coverage of urea reactant via inhibiting adsorption of OH⁻ anion to obviate OER; (2) it promotes the C − N bond cleavage of urea molecules to generate more CNO⁻ intermediates for boosting UOR. As a result, the optimized sulfur oxyanion nickel (Ni-SO$_X$) sample exhibits an ultrahigh current density of 323.4 mA cm⁻² and a 99.3 ± 0.4% UOR selectivity at 1.65 V (vs. RHE). This versatile and feasible oxyanion-engineered strategy solves competitive adsorption of organic reactant and hydroxyl anion on the active sites and opens a fresh avenue for the design of high-performance electrocatalysts under large current density operations, and therefore, is expected to be extended to other organic electro-oxidation reactions proceeding in aqueous electrolyte to obviate competing OER.

## Results

### Catalyst preparation and characterization

Pre-catalyst NiS$_2$ was prepared via a sulfuration of Ni(OH)$_2$ precursors (Supplementary Fig. 1, and Methods section). High-angle annular dark-field scanning transmission electron microscopy (HAADF-STEM) and X-ray diffraction (XRD) confirmed a pure phase of NiS$_2$ (PDF: 03-065-3325) with nanoparticles morphology. Energy-dispersive X-ray (EDX) mapping images evidenced that Ni and S atoms are uniformly distributed (Fig. 1a and Supplementary Fig. 2). After an electrochemical activation at a potential of 1.45 V for 600 s in urea-containing alkaline solution (1 M KOH + 0.33 M urea, Supplementary Fig. 3), derived electrocatalyst was obtained, which exhibited a core-shell structure with a crystalline NiS$_2$ core and a 2–5 nm thick amorphous shell (Fig. 1b). EDX mapping images and line analyses showed that the Ni and O elements present mainly in the surface layer with small amount of S, evidencing that the sulfur species exist with nickel (oxy)hydroxide in the shell (Fig. 1c and Supplementary Fig. 4)[26]. The derived electrocatalyst was denoted therefore Ni-SO$_X$. Compared with fresh NiS$_2$, Ni-SO$_X$ exhibited weaker peaks for Ni-S binding and a stronger S-O peak in the X-ray photoelectron spectroscopy (XPS) patterns (Fig. 1d), demonstrating

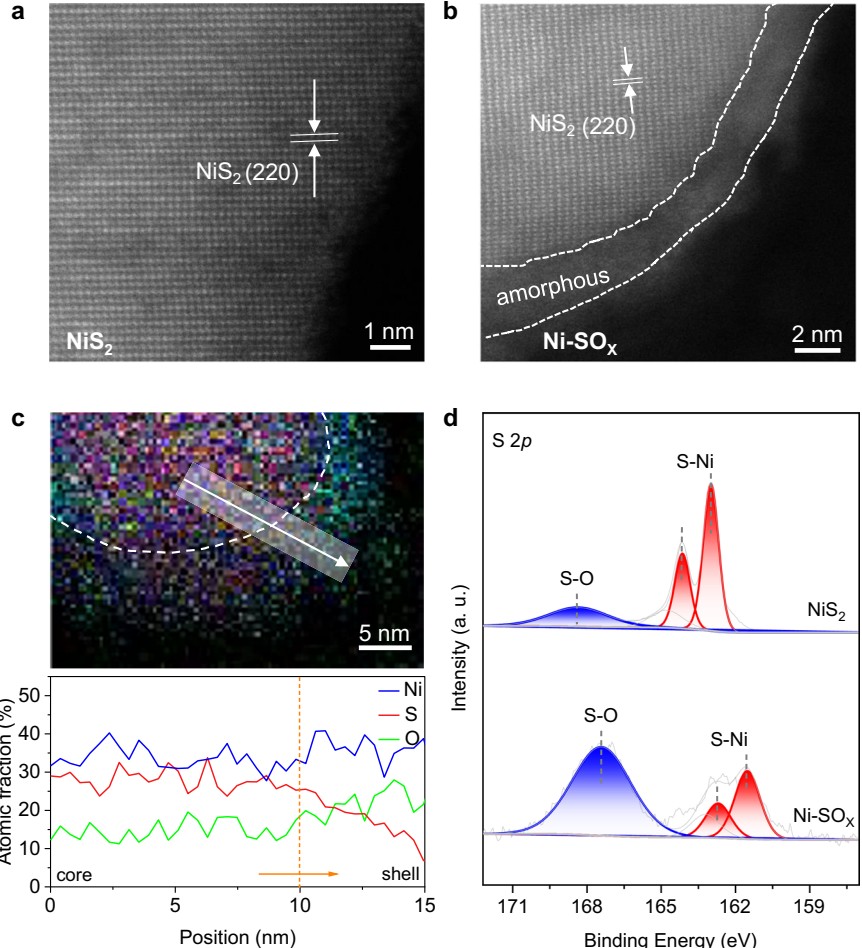

**Fig. 1 | Activation of NiS$_2$ to Ni-SO$_X$ under anodic potential.** HRTEM image of **a** NiS$_2$ and **b** Ni-SO$_X$ catalyst. **c** EDX elemental mapping (upper) and linear scan (lower) for Ni-SO$_X$. **d** S 2$p$ XPS curves for NiS$_2$ and Ni-SO$_X$ catalyst.

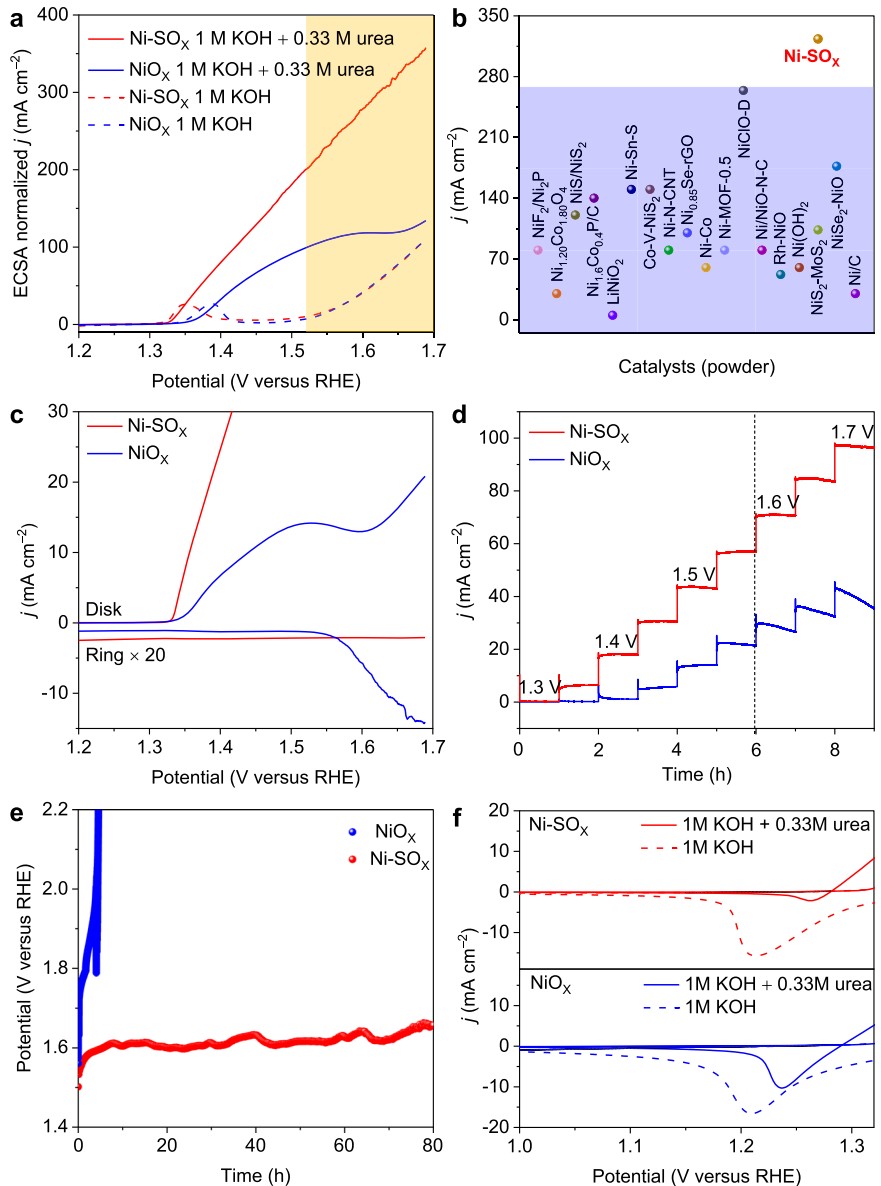

**Fig. 2 | Electrocatalytic UOR performance for Ni-SO$_X$ and NiO$_X$. a** ECSA-normalized LSV curves in 1 M KOH solution with (solid line), or without (dash line) 0.33 M urea. **b** Comparison of current density on Ni-SO$_X$ with reported powder catalysts under a 1.50 – 1.65 V potential zone for UOR. **c** In situ evaluation of O$_2$ on Ni-SO$_X$ and NiO$_X$ using RRDE in 1 M KOH with 0.33 M urea. **d** Multi-step chronopotentiometric test at different potentials from 1.30 to 1.70 V. **e** Long-term chronopotentiometric stability determined at current density 50 mA cm$^{-2}$. **f** Comparison of Ni$^{3+}$ reduction peaks for Ni-SO$_X$ and NiO$_X$ in 1 M KOH with (solid line) or without (dash line) 0.33 M urea.

that partial S in the matrix was oxidized to sulfate interacting with nickel (oxy)hydroxide in the shell of Ni-SO$_X$[26,27]. This finding is also confirmed by the Near-edge X-ray absorption fine structure spectroscopy (NEXAFS) for S K-edge, in which Ni-SO$_X$ exhibited a weaker peak for S$_2^{2-}$ and a stronger peak for SO$_4^{2-}$ compared with NiS$_2$ (Supplementary Fig. 5)[28,29]. For comparison, NiO$_X$ was also synthesized *via* electrooxidation of pristine Ni(OH)$_2$ catalysts without sulfuration, which presented typical nickel (oxy)hydroxide on the surface shell (Supplementary Figs. 6 and 7).

## Electrocatalytic UOR performance

UOR performance for Ni-SO$_X$ and NiO$_X$ powder electrodes were tested in 1 M KOH with 0.33 M urea solution *via* a typical three-electrode system. Both the Ag/AgCl (saturated KCl) with a salt bridge and Hg/HgO were used as the reference electrodes. As is shown in the electrochemically active surface area (ECSA) normalized linear sweep

voltammetry (LSV) curve (Fig. 2a and Supplementary Figs. 8, 9), both Ni-SO$_X$ and NiO$_X$ exhibit higher anodic current toward UOR (solid line) than that for OER (dash line). Overall OER performance for Ni-SO$_X$ and NiO$_X$ is similar, however UOR performance for the former is significantly better than that for the latter (confirmed by a smaller Tafel curves in Supplementary Fig. 10). At 1.65 V, UOR current density on Ni-SO$_X$ is as high as 323.4 mA cm$^{-2}$, which is *ca*. 3 times greater than that on NiO$_X$. Significantly, this finding is amongst 'best' reported performance for powder Ni-based UOR electrocatalysts (Fig. 2b and Supplementary Table 1). To exclude the effect of different nanostructures on the final UOR performance, we also synthesized Ni(OH)$_2$* nanoparticles precursors with similar morphology to NiS$_2$ nanoparticles precursors (Supplementary Fig. 11). After the electrochemical activation, the ECSA-normalized UOR performance of Ni-SO$_X$ remained superior to NiO$_X$* (Supplementary Figs. 12, 13). This finding confirms that the significant UOR current density for Ni-SO$_X$ is because of the

critical roles of sulfur oxyanion dopant and, importantly, not the nanostructure of the catalyst itself.

It is noteworthy that the UOR current density for NiO$_X$ exhibits current passivation at a high potential (>1.50 V, yellow-colour zone) (Fig. 2a), while that for Ni-SO$_X$ exhibits a potential-current linear relationship. To exclude the effect of diffusion limit, we performed the polarization tests at different rotating speeds of 400, 800, and 1600 rpm in 1 M KOH with 0.33 M urea electrolyte (Supplementary Fig. 14). The LSV curves for Ni-SO$_X$ and NiO$_X$ exhibited almost constant current densities with an increasing rotating speed. This finding evidences that UOR behaviors for Ni-SO$_X$ and NiO$_X$ are diffusion-independent. Because this current passivation in NiO$_X$ is exhibited near the OER zone, we hypothesized that it is current consumption by the $4e^-$ transferred OER rather than the $6e^-$ transferred UOR. This was confirmed via a rotating ring-disk electrode (RRDE) where UOR /OER occurred on the disk and generated O$_2$ (if any) was reduced on the Pt ring electrode. As is shown in Fig. 2c, compared with the significant reduction current for NiO$_X$ under potential >1.50 V, no ring current is exhibited with Ni-SO$_X$ in the test range. This difference demonstrates that UOR selectivity on Ni-SO$_X$ was *ca.* 100%, whereas a certain amount of OER exhibits on NiO$_X$ under a high potential.

As expected, the partial OER influences the stability of the whole electrocatalyst. As is shown in Fig. 2d, the current density for NiO$_X$ decays significantly from 1.50 V, and the decay rate exceeds 20% at 1.70 V, in contrast to Ni-SO$_X$ which remains stable at all potentials. This difference is evidenced also in the long-term stability of Ni-SO$_X$ and NiO$_X$ via chronopotentiometry at 50 mA cm$^{-2}$ (Fig. 2e), in which NiO$_X$ remained stable for only 3 h, significantly less than the 80 h for Ni-SO$_X$. The performance degradation for NiO$_X$ is likely because of the generation of high valance Ni species (> +3) associated with OER (Fig. 2f). Specifically, in 1 M KOH medium, Ni-SO$_X$ and NiO$_X$ have similar high state Ni$^{3+}$ reduction peaks because of continuous oxidation potentials and OER. However, in 1 M KOH with 0.33 M urea, Ni-SO$_X$ exhibited a much smaller Ni$^{3+}$ reduction peak than NiO$_X$, evidencing that the Ni

species maintain a relatively low valance state on Ni-SO$_X$ because of UOR (we will discuss the reason in later sections), which leads to it exhibiting greater stability than NiO$_X$.

## UOR/OER selectivity quantification

In situ differential electrochemical mass spectrometry (DEMS) was used to determine the potential dependent UOR/OER selectivity on the electrocatalysts (Fig. 3a, b). For Ni-SO$_X$, from 1.31 to 1.65 V, the N$_2$ signal was detected without attenuation, with O$_2$ negligible. In comparison, the N$_2$ signal weakened and O$_2$ signal gradually increased for NiO$_X$ from 1.50 V. This finding is in good agreement with experimental results of RRDE. The potential dependent Faradic efficiency (FEs) for each ion and gaseous product from UOR and OER (if any) were quantified via ion chromatography (IC) and gas chromatography (GC) (Fig. 3c, d and Supplementary Fig. 15). Error bars indicate the standard deviation based on three independent measurements. The N-containing products from UOR are nitrite (NO$_2^-$), N$_2$, nitrate (NO$_3^-$), and cyanate (CNO$^-$), of which NO$_2^-$ is main. The generated N$_2$ may follow the equation: $CO(NH_2)_2 + OH^- \rightarrow CO_2 + N_2 + H_2O + 6e^-$, and the typical reaction pathway: urea adsorption; dehydrogenation of N − H; C − N bond breakage; N − N coupling; N$_2$ and CO$_2$ desorption[10,12,16,21]. The C-containing products from UOR are CNO$^-$ and carbonate (CO$_3^{2-}$). The FEs for UOR products are computed based on N-containing products (N$_2$, NO$_2^-$, and NO$_3^-$). For Ni-SO$_X$, the FEs for N-containing products (FE$_{N-products}$) from UOR remained above 95 ± 4% at all potentials with few OER products (O$_2$). In contrast, for NiO$_X$, the FE$_{N-products}$ monotonically decreased, and FE$_{O_2}$ increased with increasing applied potential. Specifically, UOR selectivity was reduced to 82.6 ± 0.7% at 1.65 V for NiO$_X$, whilst that for Ni-SO$_X$ remained up to 99.3 ± 0.4%. This difference in OER /UOR selectivity for NiO$_X$ and Ni-SO$_X$ leads to the disparity in the apparent current (Fig. 2a). Significantly, this UOR selectivity on Ni-SO$_X$ is one of the highest reported Ni-based electrocatalysts under large current density operation conditions (Supplementary Table 2).

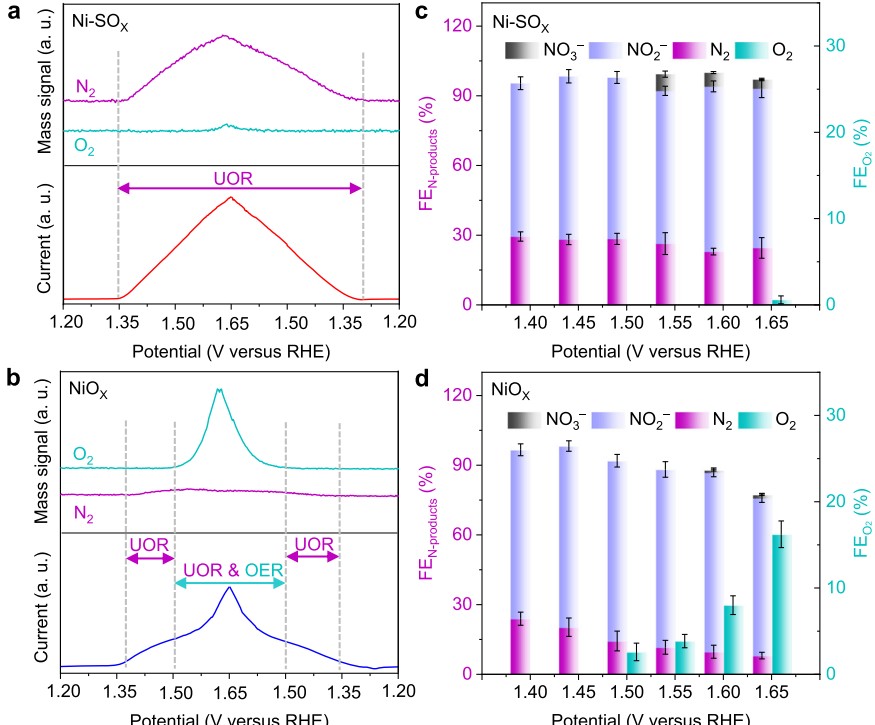

**Fig. 3 | UOR /OER selectivity quantification.** In situ detection of the representative gaseous products of UOR (N$_2$) and OER (O$_2$) via DEMS (upper) and the corresponding current acquired in 1 M KOH with 0.33 M urea (lower, without *iR* correction) on **a** Ni-SO$_X$ and **b** NiO$_X$. FEs for different UOR /OER products under different potentials on **c** Ni-SO$_X$ and **d** NiO$_X$. Error bars correspond to the standard deviation of three independent measurements.

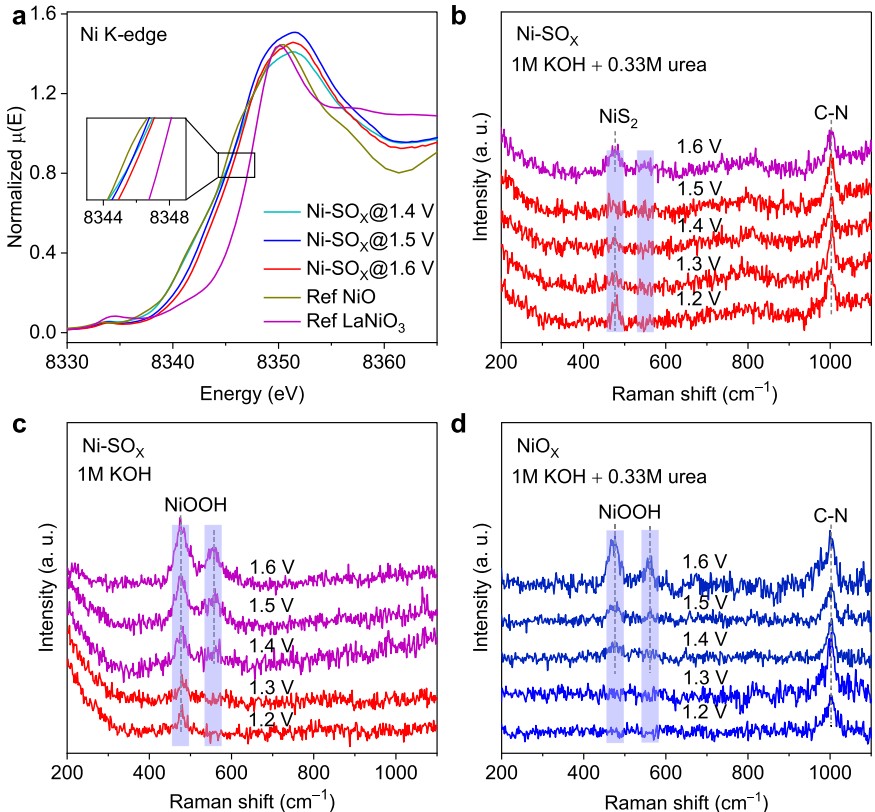

**Fig. 4 | Dynamic state change of Ni during UOR. a** In situ Ni K-edge XANES spectra for Ni-SO$_X$ under varying potential during UOR. Inset, magnified absorption edge region. In situ Raman spectra for Ni-SO$_X$ electrode in 1 M KOH **b** with, and **c** without, 0.33 M urea at applied potentials. **d** In situ Raman spectra for NiO$_X$ electrode in 1 M KOH with 0.33 M urea.

## Dynamic state change of Ni during UOR

To determine the origin for the high UOR activity, selectivity, and stability on Ni-SO$_X$, we performed in situ X-ray absorption near-edge structure (XANES) and in situ Raman spectra to gain some insights into the influence of oxyanion incorporation on active Ni sites during UOR. With applied potential increasing, the adsorption edge of Ni-SO$_X$ is positively shifted towards higher energy than that of Ref NiO, but still lower than that of Ref LaNiO$_3$, which is evidence that the state of Ni in Ni-SO$_X$ increased from +2 and remained below +3 during UOR up to 1.60 V (Fig. 4a). And a linear relationship between the Ni K-edge absorption edge and the Ni oxidation states in Ni-SO$_X$@1.4 V, Ni-SO$_X$@1.5 V, Ni-SO$_X$@1.6 V, Ref NiO, and Ref LaNiO$_3$ is plotted (Supplementary Fig. 16)[30,31]. The in situ Raman spectra for Ni-SO$_X$ show that, with 0.33 M urea in the electrolyte, there is not a high state of NiOOH accumulation (note that the peak at 475 cm$^{-1}$ is attributed to $A_g$ mode of NiS$_2$ in the core of Ni-SO$_X$[32,33] and remained almost unchanged at different voltage) (Fig. 4b). NiOOH species were detected in pure 1 M KOH electrolyte in which OER is dominated (Fig. 4c), which is in good agreement with the results from the CV curves (Fig. 2f). This can be explained that when there is urea in the electrolyte, the potential induced NiOOH rapidly catalyzes the nucleophile urea molecule to oxidation products and spontaneously reduces to Ni$^{2+}$ [21,22,34]. Conversely, accumulation of abundant NiOOH is found on NiO$_X$ in 1 M KOH with 0.33 M urea electrolyte (Fig. 4d). This evidences that NiO$_X$ has 'slow' UOR kinetics, causing accumulation of NiOOH, which then acts as active species for OER, leading to O$_2$ evolution.

## UOR mechanism analyses

In situ ATR-IR spectroscopy was used to explore how sulfur oxyanion dopant promotes UOR kinetics and inhibits OER on Ni-SO$_X$. As is shown in Fig. 5a, b, a urea's C−N stretching vibration peak at ∼1504 cm$^{-1}$ was found in both Ni-SO$_X$ and NiO$_X$ in the initial stage[35,36].

With the increase in potentials, this band on Ni-SO$_X$ decreased more than that on NiO$_X$, evidencing that the C−N bond can be more readily cleaved on Ni-SO$_X$. This is evidenced also by a stronger CNO$^-$ intermediate vibration peak at ∼2168 cm$^{-1}$ on Ni-SO$_X$[37,38]. Then, we constructed relatively matched NiOOH-SO$_4$ and NiOOH models to simulate active sites of Ni-SO$_X$ and NiO$_X$ during UOR and speculated possible pathways of urea oxidation to main product NO$_2^-$ based on the observed intermediate (Supplementary Fig. 17). The computational results show that C−N bond cleavage is promoted on NiOOH-SO$_4$ compared with NiOOH, which is consistent with in situ ATR-IR data (Supplementary Fig. 18). To further explore the origin of this C-N bond cleavage promotion on Ni-SO$_X$, the surface adsorption behaviour of the two catalysts was investigated. In the initial stage, an identical stretching mode for the O−H group in H$_2$O molecule was found for both Ni-SO$_X$ and NiO$_X$ (Fig. 5c, d)[39,40]. As the potential increased to 1.50 V, these peaks exhibited a red-shift from 3606 to 3563 cm$^{-1}$ on NiO$_X$, whilst there was no change for Ni-SO$_X$. This red-shift, together with potentials confirms generation of OH* as a surface-adsorbed species, according to the Stark tuning phenomenon[40,41]. Importantly, given that the adsorption of OH* on the active sites of catalysts is the first step in OER, this atomic level observation explains the selectivity of OER on NiO$_X$ at a high potential. However, the surface of Ni-SO$_X$ is covered by urea molecules which then readily are attacked by OH$^-$ species from the alkaline electrolyte, thereby promoting the C-N cleavage of urea for boosted UOR and obviating adsorption of OH* for suppressed OER (Fig. 5e, f). An experimental kinetics study exhibits that the reaction order of UOR for NiOOH-SO$_4$ with respect to OH$^-$ concentration is 0.65, a value significantly less than that for NiOOH of

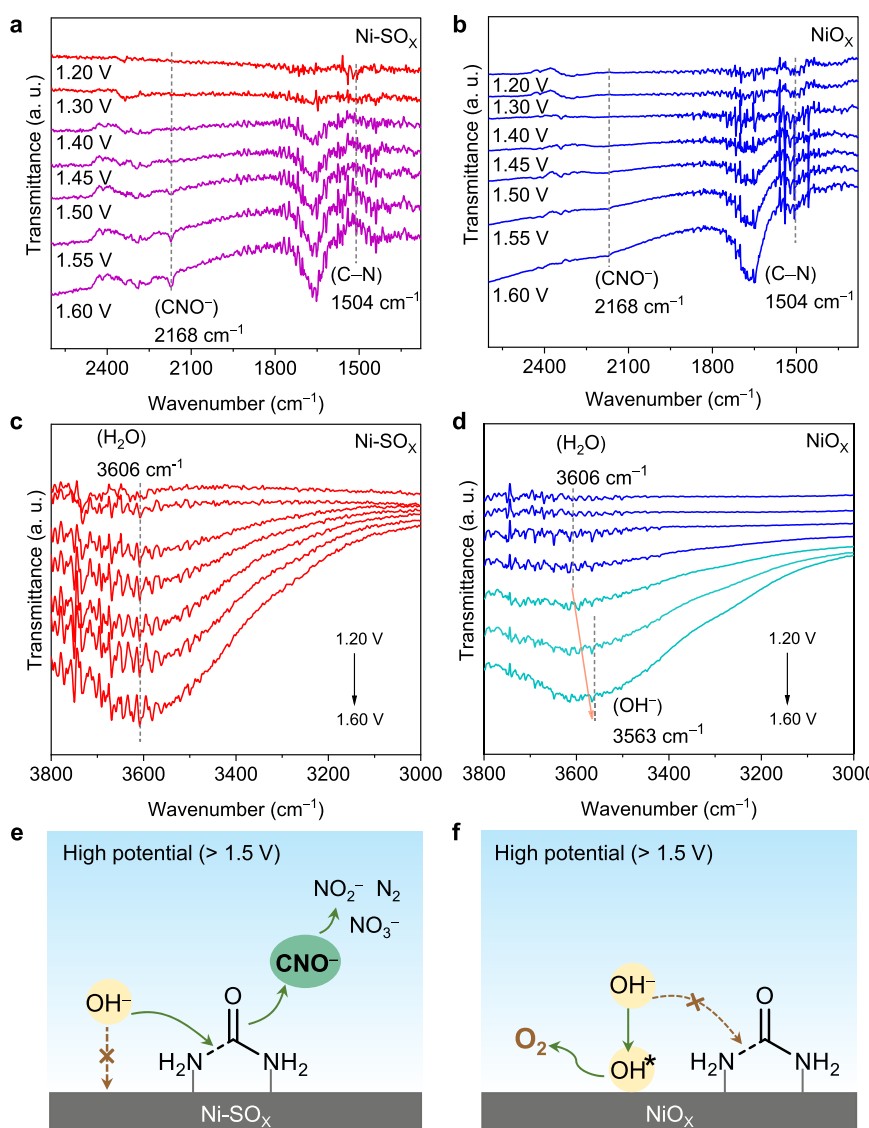

**Fig. 5 | UOR mechanism analyses. a–d** In situ ATR-IR spectra in potential window 1.20 to 1.60 V for Ni-SO$_X$ and for NiO$_X$ in different ranges. **e, f** Schematic for competing adsorption of urea and hydroxyl on Ni-SO$_X$ and NiO$_X$ surfaces.

1.87, further confirming that UOR for NiOOH-SO$_4$ exhibits a weaker dependence on OH$^-$ concentration (Supplementary Fig. 19)[8,42].

To validate the inhibiting role of oxyanion toward OH$^-$ adsorption on more Ni-based electrocatalysts, we further tested the UOR selectivity of other Ni compounds including NiSe$_2$ and Ni$_5$P$_4$ (Supplementary Fig. 20), which was derived to Ni-SeOx and Ni-POx during UOR. HAADF-STEM images and XPS spectra evidence that selenium and phosphorus oxyanion doped amorphous nickel (oxy)hydroxides are also formed on the surface of Ni-SeO$_4$ and Ni-PO$_4$ (Supplementary Figs. 21–23). As expected, selenium and phosphorus oxyanion doping boosts UOR activity and inhibits OER as sulfur (Supplementary Fig. 24).

## UOR /OER Competition Mechanism

To manipulate the competing adsorption of urea and OH$^-$, electrochemical tests were performed in electrolytes containing different concentrations of reactants. As is shown in Fig. 6a, a greater OH$^-$ concentration in electrolyte leads to more significant OER competition for NiOx, whilst having no apparent effect for Ni-SO$_X$ (Supplementary Fig. 25). The increase in urea concentration slightly attenuates but does not eliminate competition between UOR and OER on NiO$_X$ (Fig. 6b), evidencing that the surface of NiO$_X$ is mainly covered

by OH$^-$ species but not urea reactant, leading to dominant OER selectivity.

Based on above electrochemical tests and in situ spectroscopic studies, a possible UOR mechanism on two kinds of electrocatalysts including adsorption of reactants and dynamic evolution of active sites is proposed. When the applied potential is low (<1.50 V), Ni-SO$_X$ and NiO$_X$ are derived to, respectively, NiOOH-SO$_4$ and NiOOH on the surface through electrochemical oxidation (Supplementary Fig. 26). The generated NiOOH-SO$_4$ and NiOOH catalyze the nucleophile urea molecules into N products and spontaneously reduced to low-valent Ni-SO$_X$ and NiO$_X$, maintaining a good balance of derivatization and reduction. Compared with NiOOH, NiOOH-SO$_4$ makes the C − N of urea more readily cleaved, resulting in boosted UOR. Once the potential is applied over 1.50 V (Fig. 6c, d), derived NiOOH is attacked by OH$^-$ to generate high valance Ni species (≥ 3). These over-oxidized Ni species significantly participate in competing OER, which makes less NiOOH oxidizing urea, and spontaneous reduction to NiO$_X$ is slowed, leading to UOR-OER competition. However, derived NiOOH-SO$_4$ can inhibit the adsorption of OH$^-$, making it covered by sufficient nucleophile urea molecules, thereby obviating the generation of higher-valent OER-active species. Simultaneously, the repulsion of

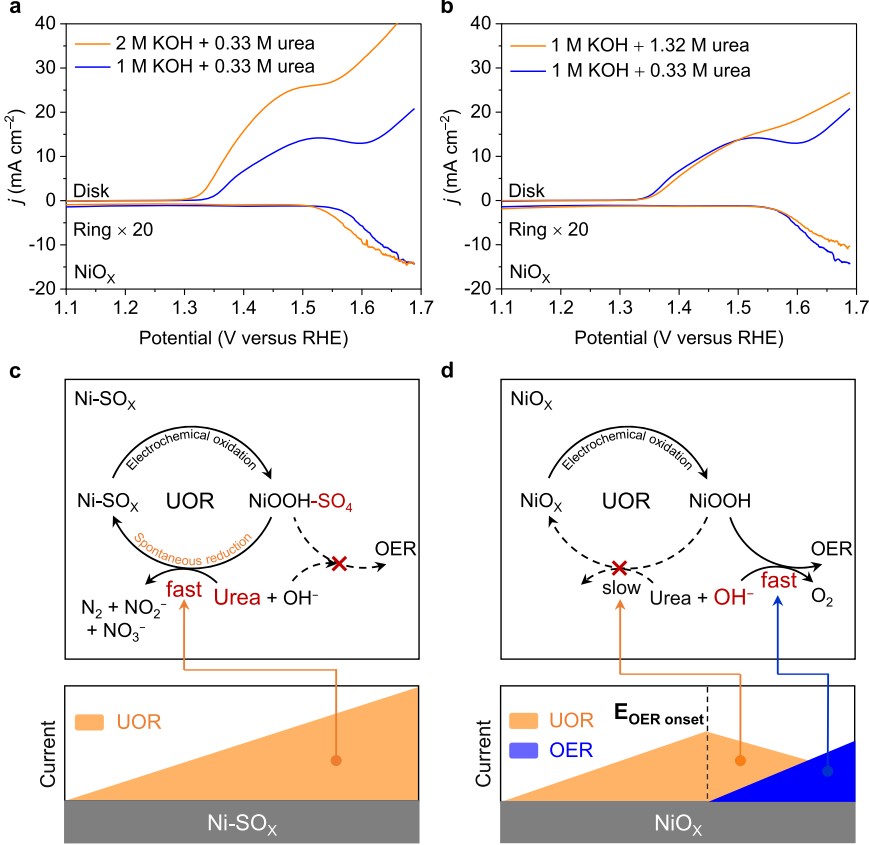

**Fig. 6 | UOR /OER competition mechanism. a** RRDE curves for NiO$_X$ obtained in different concentration of KOH with 0.33 M urea. **b** RRDE curves for NiO$_X$ obtained in 1 M KOH with different concentrations of urea. **c** Representation for UOR mechanism on Ni-SO$_X$ accompanying adsorption of reactants and dynamic

evolution of active sites under high potential (>1.50 V). **d** Representation of UOR and OER mechanism on NiO$_X$ accompanying adsorption of reactants and dynamic evolution of active sites under high potential (>1.50 V).

OH$^-$ on NiOOH-SO$_4$ promotes cleavage of C − N of urea molecules and generates more CNO$^-$ intermediates, resulting in faster UOR kinetics and more facile spontaneous reduction into Ni-SO$_X$ as a clean active site for the next derivatization and reduction cycle.

## Discussion

In summary, we achieved ultrahigh activity, selectivity, and stability of UOR *via* constructing oxyanion-engineered nickel catalysts (Ni-SO$_X$, Ni-PO$_X$, and Ni-SeO$_X$). Notably, the optimal Ni-SO$_X$ exhibited ultrahigh 323.4 mA cm$^{-2}$ UOR current density at 1.65 V with nearly 100% selectivity of N-products. The combination of diverse in situ spectroscopic measurements and DFT calculations evidenced the essential roles of oxyanion for UOR. On one hand, it inhibits the competitive adsorption of OH$^-$ on the Ni active sites to avoid OER. On the other hand, it accelerates urea's C − N bond cleavage to form CNO$^-$ intermediates for boosting UOR. Correspondingly, we proposed a comprehensive mechanism for competitive adsorption behaviour between OH$^-$ and urea to boost UOR and dynamic change of Ni active sites. We expect that this strategy will aid future research in practical urea electrolysis, and other multi-electron organic molecule oxidation coupled with cathodic hydrogen evolution for overall atomic economy and additional green energy production.

## Methods

### Synthesis of Ni(OH)$_2$, NiS$_2$, Ni$_5$P$_4$, and NiSe$_2$

In typical synthesis for Ni(OH)$_2$ catalyst, 5 mmol nickel (II) nitrate hexahydrate (Ni(NO$_3$)$_2$·6H$_2$O) and 10 mmol hexamethylenetetramine (HMT) were dissolved in 35 mL of deionized (DI) water under vigorous

stirring for 30 min to form a transparent solution. The mixture was transferred to a 50 mL Teflon-lined autoclave, sealed, and heated at 120 °C for 12 h. The resulting powder was washed with ethanol /water and dried overnight to obtain Ni(OH)$_2$ powder. NiS$_2$ catalyst was obtained from Ni(OH)$_2$ via a simple calcination. In detail, 20 mg of Ni(OH)$_2$ and 0.8 g sulfur (S) powder were placed in two separate porcelain boats and put into a tube-furnace, with S powder on the upstream. After flushing with Ar for ~30 min, the sample was heated to 350 °C at a rate 2 °C min$^{-1}$ in Ar for 2 h and cooled naturally to room temperature (RT). The Ni$_5$P$_4$ catalyst was obtained via replacing 0.8 g S with 0.8 g hypophosphite monohydrate (NaH$_2$PO$_2$•H$_2$O). The NiSe$_2$ catalyst was obtained via replacing 0.8 g S with 0.4 g Selenium (Se) powder heated to 450 °C at a rate 20 °C min$^{-1}$ in Ar for 1 h.

### Synthesis of NiO$_X$, Ni-SO$_X$, Ni-PO$_X$ and Ni-SeO$_x$

NiO$_X$, Ni-SO$_X$, Ni-PO$_X$ and Ni-SeO$_4$ catalysts were prepared via electrochemical activation of Ni(OH)$_2$, NiS$_2$, Ni$_5$P$_4$ and NiSe$_2$ at a potential of 1.45 V for 600 s in 1 M KOH with 0.33 M urea aqueous solution.

### Characterizations

X-Ray Powder Diffraction (XRD) data were collected on a Rigaku MiniFlex 600 X-Ray Diffractometer. HAADF-STEM images were obtained using an FEI Titan G2 80–300 microscope at 300 kV equipped with a probe corrector. EDX imaging was carried out with an FEI Titan Themis 80–200 microscope equipped with a SuperX detector. X-ray photoelectron spectroscopy (XPS) analyses were conducted under ultra-high vacuum on a Kratos Axis Ultra with a Delay Line Detector photoelectron spectrometer using an aluminum

monochromatic X-ray source. XPS data were corrected using the C1*s* line at 284.8 eV. In situ XAS data were collected on XAS beamline of Australian Synchrotron (ANSTO, Melbourne) using a home-made cell and processed via the Athena program. In situ Raman spectroscopy data were obtained using a Via-Reflex spectrometer (Renishaw) with a laser excitation wavelength of 532 nm. The measured potential was in the range of 1.20 to 1.60 V *vs*. RHE controlled by a CHI 760E electrochemical workstation.

### In situ DEMS measurement
A 60 nm gold-sputtered PTFE membrane was used as working electrode substrate. Electrocatalyst ink was added onto the gold-sputtered membrane and dried at RT to a uniform layer. Real-time generated gaseous products ($N_2$ and $O_2$) during the Cyclic voltammetry (CV) test were pumped to in situ DEMS system (HPR-40, Hiden Analytical Ltd.), and the signal with mass-to-charge ratio of 28, 32, was obtained. CV test was performed at a rate 10 mV s$^{-1}$ under 1 M KOH with 0.33 M urea.

### In situ Attenuated Total Reflectance Infrared (ATR-IR) spectroscopy measurement
In situ ATR-IR measurements were performed on a Thermo-Fisher Nicolet iS20 equipped with a liquid nitrogen-cooled HgCdTe (MCT) detector and a VeeMax III ATR accessory (Pike Technologies). A silicon prism coated with Au film (60°, PIKE Technologies) was mounted in a PIKE electrochemical three-electrode cell with an Ag/AgCl reference electrode (Pine Research) and a Pt counter electrode. Catalysts were sprayed onto the Au film as a working electrode. During testing, the electrolyte (1 M KOH with 0.33 M urea solution) was purged with Ar continuously. A CHI 760E electrochemical workstation was used to perform chronoamperometric tests from 1.20 to 1.60 V vs. RHE. ATR-IR curves were concurrently collected with 64 scans and a spectral resolution of 4 cm$^{-1}$.

### Electrochemical measurement
Electrochemical data were recorded via a CHI 760E electrochemical workstation. Tests were conducted in a gas-tight three-electrode H-cell at 25 °C (with Ar purge). Anode and cathode compartments were separated by a proton exchange membrane (Nafion 117). For the working electrode, 3.5 mg catalyst, 1 mg carbon black, and 10 μL 5 wt% Nafion solution were dispersed in 400 μL ethanol. Catalyst dispersion was dropped on a carbon-fibre paper or a rotating disk electrode as working electrode ($1 \times 0.5$ cm$^{-2}$, loading mass: ~0.5 mg cm$^{-2}$). The Ag/AgCl (saturated KCl) with a salt bridge and Hg/HgO were used as the reference electrodes, while a graphite rod was used as the counter electrode. The electrolyte was 1 M KOH (or 2 M KOH, 4 M KOH) with and without 0.33 M (or 1.32 M) urea. The linear sweep voltammetry (LSV) and cyclic voltammetry (CV) tests were collected at a scan rate of 5 mV s$^{-1}$ with Ar purging. The estimated electrochemical double layer capacitances ($C_{dl}$) were obtained via CV testing in a non-Faradaic potential region, 1.09 ~ 1.17 V *vs*. RHE at a scan rate of 20, 40, 60, 80, and 100 mV s$^{-1}$. Potentials were converted to reversible hydrogen electrode (RHE) and corrected with 90% *iR*-compensation, except however for stability experiments.

### Product analyses
The gas products ($N_2$, $O_2$) formed during urea oxidation were analyzed every 15 minutes via on-line gas chromatograph (GC, 8890, Agilent). The GC was fitted with HP-PLOT Q and CP-Molsieve 5 Å PT column (both Agilent), together with thermal conductivity (TCD) and flame ionization (FID) detectors. Liquid nitrate and nitrite ($NO_3^-$ and $NO_2^-$) were determined via ion chromatography (IC, Thermo Scientific Dionex Integrion RFIC) equipped with a Dionex IonPac AS19-4 μm $2 \times 250$ mm column. Calibration-curve standards, containing 5–200 ppm $NO_3^-$/$NO_2^-$ were prepared from commercially available solution of the anions (1000 ppm, Sigma-Aldrich). FE for the formation of the product was computed from:

$$FE = eF \times n/Q = eF \times n/(I \times t) \tag{1}$$

in which e is the number of transferred electrons for each product, F Faraday constant, Q charge, I applied current, t reaction time, and n, total product (mole).

Measurements for all products were repeated at least 3 times during each applied potential to exclude any possible errors.

### Reporting summary
Further information on research design is available in the Nature Portfolio Reporting Summary linked to this article.

## Data availability
Data that support findings from this study are available from the corresponding author upon reasonable request. The source data underlying Figs. 1–6 are provided as a Source Data file. Source data are provided in this paper.

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

## Acknowledgements

We gratefully acknowledge the financial support provided by the Australian Research Council through the Discovery Project and Linkage Project Programs [FL170100154, FT200100062, DP220102596, DP190103472, and LP210301397 (S.-Z.Q.)]. In situ XAS was undertaken on the X-ray Absorption Spectroscopy beamlines at the Australian Synchrotron, ANSTO, Melbourne. X.G. was supported by the Chinese CSC Scholarship Program. DFT computations were undertaken with the support of supercomputing resources provided by the Phoenix HPC service at The University of Adelaide.

## Author contributions

Y.Z. and S.-Z.Q. conceived and supervised the research. X.G. and P.W. designed and conducted the experiments. X.G., Y.Z., and S.-Z.Q. performed data analyses. X.B. and Y.J. performed the DFT computations. X.G., Y.Z., K.D., and S.-Z.Q. wrote the paper. All authors discussed the results and commented on the manuscript.

## Competing interests

The authors declare no competing interests.
