## [Peer Review File · Nature Communications]

Boosting Urea Electrooxidation on Oxyanion-Engineered Nickel Sites via Inhibited Water OxidationREVIEWER COMMENTS

Reviewer #1 (Remarks to the Author):

In this paper, the authors focused on urea electrooxidation reaction (UOR) and developed novel oxyanion-engineered nickel catalysts. The optimized sulfur oxyanion nickel (Ni-SOX) successfully suppressed the competing oxygen evolution reaction (OER) during UOR and achieved an ultrahigh current density with nearly 100 % UOR selectivity. Moreover, the authors revealed the key roles of oxyanion dopant in UOR and elucidated comprehensive UOR mechanism through a series of in situ characterization techniques. This is a well-organized and solid work, which will guide the design of high-performance electrocatalysts for small organic electrooxidation reactions. Therefore, I strongly recommend its publication in Nature Communications with minor revision.

1. In Figure 3c-d, the potential dependent Faradic efficiency for ion and gaseous products were quantified via ion chromatography and gas chromatography, respectively. The authors should provide more measurement details (e.g., calibration curves).
2. According to the in situ Raman results, the authors claim that the NiOOH species detected on NiOX in 1 M KOH electrolyte with 0.33 M urea acts as active species for OER but lack of comparative experiment. Thus, in situ Raman measurement should be conducted for NiOX in 1M KOH electrolyte without urea for comparison.
3. Tafel plot is often useful to reflect the catalytic performance of materials. Therefore, it is necessary to compare the Tafel slope for Ni-SOX and NiOX, respectively, to visually reflect the ability of the as-prepared catalysts for urea electrooxidation.
4. The XRD pattern and HRTEM image of NiSe₂ and Ni₅P₄ electrocatalysts should be given to help the reader confirm the phase of these catalysts.

Reviewer #2 (Remarks to the Author):

In this article, the authors investigate the ability of a Ni-derived electrocatalyst in performing the electro-oxidation of urea.

The electrocatalyst is derived from NiS₂. They authors claim that sulfate groups can be found at the surface of the electrocatalyst. I am not fully convinced.

It seems that in the case of Ni-SO_x, the diffusion limit to oxidize ie never reached. This point would deserve a clear explanation. Some characterization of the nanostructuring are missing comparing the Ni-SO_x catalyst with the fresh Ni(OH)₂ precursor and the NiOO_x catalyst. This may explain as well the good performance.

The products of reaction of electrooxidation should be full determined and explained. Generating N₂ in oxidation conditions seems surprising. In which product the urea's C end?

Moving to the DFT calculations, I have even stronger concerns related to the model used and the related conclusions. The Ni-SO_x model consists in removing a Ni and replace it by a S. Why ?? I would expect the sulfur to replace the oxygen, not the Ni cation, and then, S

would affect the redox properties of the neighboring Ni. Related conclusions are very likely misleading.

Besides, the adsorption energy of OH* is presented in the text as the adsorption of OH- which is not corresponding to the computational details (it is closer to the oxidative adsorption of water). Instead of using H₂, they should use the CHE model to properly include the influence of the potential on this adsorption. Last, the inclusion of a solvent model is not described but necessary to properly include the effect of potential and charge separation at the electrochemical interface.

For all those reasons I believe that this article does not reach the expected standards of Nature Communications.

In the meanwhile, REFEREE #3 recommend authors to have below points addressed:

This referee summarize the main achievement of the manuscript as "reported the oxyanion-engineered nickel sites enhancing the urea electrooxidation reaction (UOR) activity via inhibited water oxidation (i.e. oxygen evolution reaction; OER)." This referee thinks the novelty is not sufficient to be publishable in Nature Communications. Their further technical comments are listed as below:

1. This referee suggest reference electrode calibration must be performed on the same electrolytes to avoid any errors in UOR potential measurements. Ag/AgCl electrodes are oxidized and unstable in alkaline electrolyte and changes potential or current during UOR measurements. Thus this referee advise the author to use the reference electrode Hg/HgO or SCE in alkaline solution.
2. To avoid the error in the quantification of UOR/OER selectivity, this referee suggest authors repeat the experiments at least three times and show the data in the error bars (the standard deviation of a data set; Fig. 3,c,d).
3. The white-line area of NiO is significantly smaller than that of Ni-SO_x (Fig. 4a). Then, the author stated that "with applied potential increasing, the state of Ni increased from +2 and remained below +3 during UOR up to 1.60 V". This referee recommend authors should measure the white line area and plot the graph between the white line area (similar energy ranges) and the oxidation states at different potentials (see ref. Applied Catalysis B: Environmental, 111–112, 2012, 509-514).

Response to Reviewer #1

Reviewer's Remarks to Authors

In this paper, the authors focused on urea electrooxidation reaction (UOR) and developed novel oxyanion-engineered nickel catalysts. The optimized sulfur oxyanion nickel (Ni-SO_x) successfully suppressed the competing oxygen evolution reaction (OER) during UOR and achieved an ultrahigh current density with nearly 100 % UOR selectivity. Moreover, the authors revealed the key roles of oxyanion dopant in UOR and elucidated comprehensive UOR mechanism through a series of in situ characterization techniques. This is a well-organized and solid work, which will guide the design of high-performance electrocatalysts for small organic electrooxidation reactions. Therefore, I strongly recommend its publication in Nature Communications with minor revision.

Response

We thank Reviewer #1 for his/her valuable comments and positive recommendation for publication.

Comment 1-1

In Figure 3c-d, the potential dependent Faradic efficiency for ion and gaseous products were quantified via ion chromatography and gas chromatography, respectively. The authors should provide more measurement details (e.g., calibration curves).

Response

We agree with Reviewer #1. Following the suggestion, we added more measurement details in revised manuscript (R-MS) and revised-supporting information (R-SI) included additional explanatory detail as follows:

1) R-MS (Page 8, 9), we added the following explanatory text:

'The potential dependent Faradic efficiency (FEs) for each ion and gaseous product from UOR and OER (if any) were quantified via ion chromatography (IC) and gas chromatography (GC) (Fig. 3c, d and Supplementary Fig. 15). Error bars indicate the standard deviation based on three independent measurements. The N-containing products from UOR are nitrite (NO₂⁻), N₂, nitrate (NO₃⁻), and cyanate (CNO⁻), of which NO₂⁻ is main. The C-containing products from UOR are CNO⁻ and carbonate (CO₃²⁻). The FEs for UOR products are computed based on N-containing products (N₂, NO₂⁻, and NO₃⁻).'

2) R-SI (Page 9), we added **Fig. R1** as **Supplementary Fig. 15**:

Fig. R1 (a) GC trace for gaseous products during electrochemical UOR 1.65 V on Ni-SO_x. (b) IC trace for ion products during electrochemical UOR at 1.65 V on Ni-SO_x. Calibration curves for (c) NO₂⁻ and, (d) NO₃⁻ via IC measurement.

Comment 1-2

According to the in situ Raman results, the authors claim that the NiOOH species detected on NiO_x in 1 M KOH electrolyte with 0.33 M urea acts as active species for OER but lack of comparative experiment. Thus, in situ Raman measurement should be conducted for NiO_x in 1M KOH electrolyte without urea for comparison.

Response

In the Raman spectra, two peaks at 474 and 551 cm⁻¹ are attributed to NiOOH species. When increasing potentials during OER, these two peaks gradually appear because of accumulation of NiOOH species. This phenomenon for nickel-based catalysts during OER has been demonstrated in a number of studies including, *Nat. Commun.* 2022, 13, 6094, *J. Am. Chem. Soc.* 2013, 135, 12329-12337, and *Angew. Chem. Int. Ed.* 2019, 58, 1252-1265. Our NiO_x catalyst similarly exhibited peaks for NiOOH species in 1 M KOH electrolyte, evidencing that the derived NiOOH species detected on NiO_x in 1 M KOH electrolyte are active species for OER (**Fig. R2**).

Fig. R2 In situ Raman spectra for NiO_x electrode in 1 M KOH electrolyte.

Comment 1-3

Tafel plot is often useful to reflect the catalytic performance of materials. Therefore, it is necessary to compare the Tafel slope for Ni-SO_x and NiO_x, respectively, to visually reflect the ability of the as-prepared catalysts for urea electrooxidation.

Response

We agree. We analyzed the Tafel plot for Ni-SO_x and NiO_x. As is seen in **Fig. R3**, the Tafel slope for Ni-SO_x is calculated as 18.6 mV dec⁻¹, which is less than that for NiO_x (38.8 mV dec⁻¹). This finding evidence better UOR performance for Ni-SO_x.

Fig. R3 Tafel slops for Ni-SO_x and NiO_x catalyst during UOR.

In response to address fully this comment, **Fig. R3** has been added in R-SI (Page 7) as **Supplementary Fig. 10** with the corresponding discussion added in R-MS (Page 6):

'Overall OER performance for Ni-SO_x and NiO_x is similar, however UOR performance for the former is significantly better than that for the latter (confirmed by a smaller Tafel curves in Supplementary Fig. 10).'

Comment 1-4

The XRD pattern and HRTEM image of NiSe₂ and Ni₅P₄ electrocatalysts should be given to help the reader confirm the phase of these catalysts.

Response

X-ray diffraction (XRD) confirms the pure phase of synthesized NiSe₂ (PDF#03-065-5016) and Ni₅P₄ (PDF#00-018-0883) (**Fig. R4a, c**). High-resolution TEM (HRTEM) analysis evidences that both synthesized NiSe₂ and Ni₅P₄ exhibit nanoparticles morphology (**Fig. R4b, d**).

Fig. R4 (a) XRD pattern and (b) HRTEM image, of NiSe₂. (c) XRD pattern and (d) HRTEM image, of Ni₅P₄.

In response to address fully this comment, **Fig. R4** has been added in R-SI (Page 12) as **Supplementary Fig. 21** with explanatory text added in R-MS (Page 15):

'To validate the inhibiting role of oxyanion toward OH⁻ adsorption on more Ni-based electrocatalysts, we further tested the UOR selectivity of other Ni compounds including NiSe₂ and Ni₅P₄ (Supplementary Fig. 21), which was derived to Ni-SeO_x and Ni-PO_x during UOR.'

Response to Reviewer #2

Reviewer's Remarks to Authors

In this article, the authors investigate the ability of a Ni-derived electrocatalyst in performing the electro-oxidation of urea.

Response

We sincerely appreciate Reviewer #2 for his/her valuable comments. Accordingly, we provide more new results to address the technical issues. In a brief summary, from experimental perspective, we provide S K-edge NEXAFS spectra to evidence the exist of sulfate on the surface of Ni-SO_x. Moreover, we exclude the effect of diffusion limit and nanostructure on the final UOR performance *via* conducting polarization tests under different rotating speeds and ECSA-normalized LSV curves. From the perspective of calculations, following the suggestion, we construct the model in which S substitute O atoms (NiOOH-SO₄ (O)) and provide the calculation details of solvent model and the adsorption energy of OH*. Specific responses to the comments and corresponding modifications are provided below.

Comment 2-1

The electrocatalyst is derived from NiS₂. They authors claim that sulfate groups can be found at the surface of the electrocatalyst. I am not fully convinced.

Response

As can be seen from original Fig. 1 and Supplementary Fig. 2, Ni and S atoms in NiS₂ are uniformly distributed without any amorphous shell. However, the derived Ni-SO_x exhibited a core-shell structure with a crystalline NiS₂ core and amorphous shell. The amorphous shell exhibited a certain content of S elements and increased O elements, conforming the existence of sulfur species on the surface of Ni-SO_x. Moreover, the derived Ni-SO_x exhibited weaker peaks for Ni-S binding and a stronger S-O peak in the surface sensitive XPS patterns compared with NiS₂, confirming that partial S was oxidized to sulfate interacting with nickel (oxy)hydroxide on the surface of Ni-SO_x (similar result has also been observed in *Nat. Commun.* 2022, 13, 2916).

Additionally, we supply the near-edge X-ray absorption fine structure spectroscopy (NEXAFS) of S K-edge for Ni-SO_x and NiS₂. As is seen from **Fig. R5**, compared with NiS₂, Ni-SO_x exhibits a weak peak for S₂²⁻ and a stronger peak for SO₄²⁻, further evidencing that sulfate group can be found on the surface of Ni-SO_x.

Fig. R5 S K-edge NEXAFS spectra for NiS₂ and Ni-SO_x catalysts.

In response to address directly this comment of Reviewer #2, **Fig. R5** has been added in R-SI (Page 4) as **Supplementary Fig. 5** with the corresponding text added in R-MS (Page 5):

'This finding is also confirmed by the Near-edge X-ray absorption fine structure spectroscopy (NEXAFS) for S K-edge, in which Ni-SO_x exhibited a weaker peak for S₂²⁻ and a stronger peak for SO₄²⁻ compared with NiS₂ (Supplementary Fig. 5)²⁸⁻²⁹.'

Comment 2-2

It seems that in the case of Ni-SO_x, the diffusion limit to oxidize is never reached. This point would deserve a clear explanation.

Response

We thank Reviewer #2 for providing this critical comment. To study the effect of diffusion, we performed the polarization tests at different rotating speeds of 400, 800, and 1600 rpm on a rotating disk electrode in 1 M KOH with 0.33 M urea electrolyte. As is seen in **Fig. R6**, Ni-SO_x exhibits almost constant current densities with increasing rotating speed. This finding evidences that UOR behaviors for Ni-SO_x and NiO_x are diffusion-independent and that current passivation in NiO_x is because of competing OER.

In response to address fully this comment of Reviewer #2, **Fig. R6** has been added in R-SI (Page 8) as **Supplementary Fig. 14** with the corresponding discussion presented in R-MS (Page 6):

'To exclude the effect of diffusion limit, we performed the polarization tests at different rotating speeds of 400, 800, and 1600 rpm in 1 M KOH with 0.33 M urea electrolyte (Supplementary Fig. 14). The LSV

curves for Ni-SO_x and NiO_x exhibited almost constant current densities with an increasing rotating speed. This finding evidences that UOR behaviors for Ni-SO_x and NiO_x are diffusion-independent.'

Fig. R6 LSV curves for Ni-SO_x and NiO_x on a rotating disk electrode (RDE) at different rotation rates (without iR correction).

Comment 2-3

Some characterization of the nano structuration are missing comparing the Ni-SO_x catalyst with the fresh Ni(OH)₂ precursor and the NiO_x catalyst. This may explain as well the good performance.

Response

Following the kind suggestion of Reviewer #2, we provide a comparison of nanostructures between Ni-SO_x with Ni(OH)₂ precursor and NiO_x. Specifically, Ni-SO_x exhibited core-shell structure nanoparticles morphology (original Fig.1b and supplementary Fig. 4), whereas Ni(OH)₂ exhibited nanosheets morphology and NiO_x exhibited core-shell nanosheets morphology (original supplementary Fig. 1 and **Fig. R7**). To exclude the effect of different nanostructures on boosted UOR, we evaluated the electrochemically active surface area (ECSA) of Ni-SO_x and NiO_x and normalized the UOR performance. The ECSA for Ni-SO_x and NiO_x were obtained *via* double-layer capacitance (C_{dl}) measurement. The C_{dl} for Ni-SO_x and NiO_x were computed to be 0.82 mF cm⁻² and 0.32 mF cm⁻² (original Supplementary Fig. 8b, c). Then, the ECSA-normalized linear sweep voltammetry (LSV) curves are presented in original Fig. 2a, in which the normalized UOR current density for Ni-SO_x is still significantly greater than for NiO_x. This finding confirms that the good UOR performance for Ni-SO_x is not related to the nanostructure of the catalyst itself but depends on the critical roles of sulfur oxyanion dopant.

Fig. R7 (a) HRTEM image of NiO_x, (b) Corresponding FFT pattern selected from region b in Fig. R7a. (c) Corresponding FFT pattern selected from region c in Fig. R7a.

Additionally, We deliberately prepared another Ni(OH)₂* with similar nanoparticles morphology to NiS₂ (**Fig. R8**). After the electrochemical activation, NiO_x* exhibited similar core-shell structured nanoparticles morphology as Ni-SO_x (**Fig. R9**). The C_{dl} for NiO_x* is 0.44 mF cm⁻², and ECSA-normalized UOR performance for NiO_x* is still lower than Ni-SO_x (**Fig. R10**). This finding further confirms that the nanostructure of these catalysts is not the crucial factor for boosting UOR performance.

Fig. R8 (a) XRD pattern, (b) and (c) HRTEM image images of Ni(OH)₂*.

Fig. R9 (a) HRTEM image of NiO_x*, (b) corresponding FFT pattern selected from region b in Fig. R9a. (c) corresponding FFT pattern selected from region c in Fig. R9a.

Fig. R10 (a) Electrochemical double-layer capacitance (C_{dl}) for NiO_x^* . Inset, CV curve at different scan rates in non-Faradaic capacitance current range. (b) ECSA-normalized LSV curves for $Ni-SO_x$, NiO_x and NiO_x^* in 1 M KOH solution with (solid line), or without (dash line) 0.33 M urea.

In response to address directly this comment of Reviewer #2, **Fig. R7-10** has been added in R-SI as **Supplementary Fig. 7, 11-13** with the corresponding discussion presented in R-MS (Page 5, 6).

'For comparison, NiO_x was also synthesized via electrooxidation of pristine $Ni(OH)_2$ catalysts without sulfuration, which presented typical nickel (oxy)hydroxide on the surface shell (Supplementary Fig. 6, 7).'

'To exclude the effect of different nanostructures on the final UOR performance, we also synthesized $Ni(OH)_2^$ nanoparticles precursors with similar morphology to NiS_2 nanoparticles precursors (Supplementary Fig. 11). After the electrochemical activation, the ECSA-normalized UOR performance of $Ni-SO_x$ remained superior to NiO_x^* (Supplementary Fig. 12, 13). This finding confirms that the significant UOR current density for $Ni-SO_x$ is because of the critical roles of sulfur oxyanion dopant and, importantly, not the nanostructure of the catalyst itself.'*

Comment 2-4

The products of reaction of electrooxidation should be full determined and explained. Generating N_2 in oxidation conditions seems surprising. In which product the urea's C end?

Response

The gas products for our $Ni-SO_x$ catalyst during UOR were detected *via in situ* differential electrochemical mass spectrometry (DEMS) and gas chromatography (GC) (original Fig. 3 a, c, and **Fig. R11a**). N_2 can be detected at from 1.31 V to 1.65 V with few O_2 (FE <1%), and the Faradic efficiency

for N_2 (FE_{N_2}) is stable at $20\%-30 \pm 2\%$. Ion products were analyzed *via* ion chromatography (IC) (**Fig. R11b-d**). There are four peaks, which are assigned to nitrite (NO_2^-), cyanate (CNO^-), nitrate (NO_3^-), and carbonate (CO_3^{2-}). Amongst these, CNO^- and CO_3^{2-} are the end products of urea's C. In contrast to NO_2^- and NO_3^- , the oxidation state of carbon in CNO^- and CO_3^{2-} is the same as that in urea (+4 valance). Therefore, the FE for UOR liquid products is computed based only on NO_2^- and NO_3^- . From 1.40 V to 1.65 V, $FE_{NO_2^-}$ is stable at $65 \pm 2\%-71 \pm 2\%$, and $FE_{NO_3^-}$ is less than $10 \pm 1\%$.

Fig. R11 (a) GC trace for gaseous products during electrochemical UOR 1.65 V on Ni-SO_x. (b) IC trace for ion products during electrochemical UOR at 1.65 V on Ni-SO_x. Calibration curves for (c) NO_2^- and, (d) NO_3^- *via* IC measurement.

In response to fully address this comment, **Fig. R11** has been added in R-SI (Page 9) as **Supplementary Fig. 15** with the corresponding discussion added in R-MS (Page 8,9):

'The potential dependent Faradic efficiency (FEs) for each ion and gaseous product from UOR and OER (if any) were quantified via ion chromatography (IC) and gas chromatography (GC) (Fig. 3c, d and Supplementary Fig. 15). The N-containing products from UOR are nitrite (NO_2^-), N_2 , nitrate (NO_3^-), and cyanate (CNO^-), of which NO_2^- is main. The C-containing products from UOR are CNO^- and carbonate (CO_3^{2-}). The FEs for UOR products are computed based on N-containing products (N_2 , NO_2^- , and NO_3^-).'

Comment 2-5

Moving to the DFT calculations, I have even stronger concerns related to the model used and the related conclusions. The Ni-SO_x model consists in removing a Ni and replace it by a S. Why ?? I would expect the sulfur to replace the oxygen, not the Ni cation, and then, S would affect the redox properties of the neighboring Ni. Related conclusions are very likely misleading.

Response

We thank Reviewer #2 for pointing out this aspect about model suitability. Accordingly, we construct a new model in which S substitute O atom (NiOOH-SO₄ (O)) and introduce four additional O atoms coordinated to S as suggested (Fig. R12a). After geometry optimization, the -SO₃ group adsorbs to the top site of the O atom to form the -SO₄ group (Fig. R12b). In this structure, all Ni atoms are saturated with six O atoms in coordination, resulting in the surface O atom being the active site. The adsorption strengths of OH* on NiOOH-SO₄ (O) are then evaluated (Fig. R13). Without an applied potential, the adsorption strength of OH* on NiOOH-SO₄ (O) remains weaker than that for NiOOH. With an increase of positive potential, the adsorption strength for OH* on NiOOH-SO₄ (O) exhibits minimal change, which is consistent with the original model where S substitute Ni atoms (NiOOH-SO₄ (Ni)). The differential charge density shows the charge accumulation on sulfur oxyanion groups of NiOOH-SO₄ (O), which is in agreement with NiOOH-SO₄ (Ni) (Fig. R14). Therefore, both NiOOH-SO₄ (O) and NiOOH-SO₄ (Ni) models exhibit capability in inhibiting OH⁻ adsorption.

Fig. R12 (a) Slab model for S substitution of O atom for NiOOH-SO₄ (O). (b) Evolution of distance between S-atom in SO₄²⁻ species and H-atom layer during geometric optimization, which includes the minimum (black-color solid line), maximum (red-color solid line) and average (blue-color solid line) distances between S-atom and H-atom layer. The stable geometry is obtained when the distance reaches 4.87 Å.

Fig. R13 (a) OH* adsorbed on NiOOH-SO₄ (O). (b) Adsorption energy for OH* (ΔE_{OH^*}) on NiOOH, NiOOH-SO₄ (Ni), and NiOOH-SO₄ (O) without applied potential. (c) ΔE_{OH^*} on NiOOH-SO₄ (Ni), NiOOH-SO₄ (O) and NiOOH with varying potentials from 1.4 V to 1.8 V vs. RHE.

Fig. R14 Differential charge density for (a) NiOOH-SO₄ (Ni) and (b) NiOOH-SO₄ (O).

To further verify that the substituted O also applies to the inhibiting role of other oxyanion on OH⁻ adsorption, the effect of substitution of P and Se (NiOOH-PO₄ (O) and NiOOH-SeO₄ (O)) on the OH* adsorption strength is further evaluated. As is shown in **Fig. R15** and **Fig. R16**, the adsorption strength for OH* on NiOOH-PO₄(O) and NiOOH-SeO₄(O) is also lower than that for NiOOH and changes little with applied potential, which is consistent with the results for NiOOH-PO₄ (Ni) and NiOOH-SeO₄ (Ni). Moreover, an approximately linear correlation remains between the *p*-band center of the O atom and adsorption strength for OH* in NiOOH-SO₄ (O), NiOOH-PO₄ (O), and NiOOH-SeO₄ (O), evidencing that oxyanion doping shifts the ϵ_p down away from Fermi level to inhibit OH⁻ adsorption, thereby suppressing competition of OER.

Fig. R15 Slab model for (a) NiOOH-SeO₄(O) and (b) NiOOH-PO₄(O). OH* adsorbed (c) NiOOH-SeO₄(O) and (d) NiOOH-PO₄(O) surface.

Fig. R16 (a) Comparison of ΔE_{OH^*} on NiOOH, NiOOH-SO₄, NiOOH-SeO₄, and NiOOH-PO₄ surface with varying potentials, including S, P and Se substitution of Ni and O atoms. Projected density of state (PDOS) of the 2p band of the O atom in (b) NiOOH, NiOOH-SO₄(Ni), NiOOH-PO₄(Ni) and NiOOH-SeO₄(Ni), (c) NiOOH-SO₄(O), NiOOH-PO₄(O) and NiOOH-SeO₄(O) at 1.40 V (vs. RHE). The p-band center (ϵ_p) is marked by the gray-color solid line and the Fermi level is set as zero. (d) Relationships between ΔE_{OH^*} and ϵ_p at 1.40 V (vs. RHE) on different surfaces.

In response to fully address this comment, **Fig. R12-16** has been added in R-MS as **Fig. 6** and **Supplementary Fig. 17-20, 25, 26** with the corresponding discussion added in R-MS (Page 13, 15, 21).

'NiOOH(001) model was constructed to simulate the active sites for NiO_x in UOR /OER. Two models of S replacing O atom (NiOOH-SO₄ (O)) and S replacing Ni atom (NiOOH-SO₄ (Ni)) were established to simulate the active sites for Ni-SO_x in UOR /OER (Fig. 6 a,b and Supplementary Fig. 17, 18). Without an applied potential, the adsorption strength of OH on both NiOOH-SO₄ (O) and NiOOH-SO₄ (Ni) models is significantly lower than that of NiOOH (Supplementary Fig. 19). With positive potential increasing (Fig. 6c), the adsorption strength of OH* on NiOOH exhibits a linear increasing dependency, but that on both NiOOH-SO₄ (O) and NiOOH-SO₄ (Ni) models changes little. This finding evidences an inert OH⁻ adsorption behaviour on Ni-SO_x surface. The differential charge density shows the charge accumulation on sulfur oxyanion groups of both NiOOH-SO₄ (O) and NiOOH-SO₄ (Ni) models, indicating that the sulfur oxyanion is negatively charged (Supplementary Fig. 20).'*

'DFT calculations also show that the adsorption strength for OH on NiOOH-PO₄(O), NiOOH-PO₄(Ni), NiOOH-SeO₄(O), and NiOOH-SeO₄(Ni), are lower than that of NiOOH and change a little with applied potential (Supplementary Fig. 25, 26).'*

'Two configurations were considered for each oxygen anion, namely P, S and Se replacing the Ni and O atoms on the surface, respectively.'

Comment 2-6

Besides, the adsorption energy of OH* is presented in the text as the adsorption of OH⁻ which is not corresponding to the computational details (it is closer to the oxidative adsorption of water). Instead of using H₂, they should use the CHE model to properly include the influence of the potential on this adsorption.

Response

This is a very good question. The adsorption energy of OH* is presented in the text as the adsorption of OH⁻ because of the following reasons.

We calculate the process of OH⁻ (aq) + * → *OH + e⁻. The step is the adsorption step of OH⁻ on active site with a release of an electron:

$$\Delta G = G(\text{OH}^*) + \mu_{\text{e}^-} - (G(\text{OH}_{\text{aq}}^-) + G(*)) \quad (1)$$

Under thermodynamic equilibrium of water self-ionization, we have

$$G(\text{H}_{\text{aq}}^+) + G(\text{OH}_{\text{aq}}^-) = G(\text{H}_2\text{O}_l) \quad (2)$$

Such reaction can be that in the reversible hydrogen electrode (RHE), where the following thermodynamic equilibrium exists:

$$G(H_{aq}^+) + \mu_{e^-} = G(H_2)/2 \quad (3)$$

By substituting eq 3 into eq 2, we obtain

$$G(OH_{aq}^-) = G(H_2O_l) - G(H_2)/2 + \mu_{e^-} \quad (4)$$

By substituting eq 4 into eq 1, we obtain

$$\Delta G = G(OH^*) - G(*) - G(H_2O_l) + G(H_2)/2 \quad (5)$$

The corresponding adsorption energy is calculated as we mentioned in the manuscript:

$$\Delta E_{OH^*} = E(\text{sub}/OH) - E(\text{sub}) - E(H_2O) + 1/2E(H_2)$$

Which is in agreement with the previous study. (*Nat. Commun.* 2022, 13, 2916 and *ACS Catal.* 2019, 9, 9332–9338). Therefore, the adsorption energy of OH* is presented in our MS as the adsorption of OH⁻.

In the CHE model, the potential effects on the reaction are described implicitly via “N_eU” and the surface charge is neutral. However, the surface charge of the catalyst varies with the electrode potential under real electrochemical reaction conditions. Thus, we adopted a better approach (constant potential approach) to describes the electrochemical behaviour under real reaction conditions. We calculated the adsorption energy of OH* without potential using the CHE model as shown in **Fig. R13b**. The OH* adsorption strength still follows the order of NiOOH > NiOOH-SO₄ (O) > NiOOH-SO₄ (Ni), which is consistent with our conclusion under reaction conditions.

In response to this comment, we have added **Fig. R13b** in R-SI as **Supplementary Fig. 19** and calculation details in R-SI (Page 19):

‘Supplementary note 1: Calculation details of the adsorption energy of *OH as the adsorption of OH⁻.

*we calculate the process of OH⁻ (aq) + * → *OH + e⁻. The step is the adsorption of OH⁻ on active site with a release of an electron³⁰:*

$$\Delta G = G(OH^*) + \mu_{e^-} - (G(OH_{aq}^-) + G(*)) \quad (1)$$

Under thermodynamic equilibrium of water self-ionization, we have

$$G(H_{aq}^+) + G(OH_{aq}^-) = G(H_2O_l) \quad (2)$$

Such reaction can be that in the reversible hydrogen electrode (RHE), where the following thermodynamic equilibrium exists:

$$G(H_{aq}^+) + \mu_{e^-} = G(H_2)/2 \quad (3)$$

By substituting eq 3 into eq 2, we obtain

$$G(OH_{aq}^-) = G(H_2O_l) - G(H_2)/2 + \mu_{e^-} \quad (4)$$

By substituting eq 4 into eq 1, we obtain

$$\Delta G = G(\text{OH}^*) - G(^*) - G(\text{H}_2\text{O}) + G(\text{H}_2)/2 \quad (5)$$

The corresponding adsorption energy is calculated as we mentioned in the manuscript:

$$\Delta E_{\text{OH}^*} = E(\text{sub}/\text{OH}) - E(\text{sub}) - E(\text{H}_2\text{O}) + 1/2E(\text{H}_2)$$

Which is in agreement with the previous study^{31,32}. Therefore, the adsorption energy of OH* is presented in the text as the adsorption of OH⁻.

Comment 2-7

Last, the inclusion of a solvent model is not described but necessary to properly include the effect of potential and charge separation at the electrochemical interface.

Response

We are sorry for the confusion caused to the Reviewers. In our MS, we have used the implicit solvation model as implemented in VASPsol. The adsorption strength of OH* is evaluated under different electrode potentials. To determine the surface charge for a given electrode potential, we use the VASPsol code, which adds implicit aqueous electrolyte into the system (*J. Am. Chem. Soc.* 2021, 143, 9423–9428). We add different numbers of extra electrons ($n_{\text{extra_e}}$) and calculate the corresponding absolute Fermi levels (E_{F}) with respect to the electrostatic potential in the region far from NiOOH-TO_x (TO_x: PO_x, SO_x and SeO_x) surface. The atomic positions are fully relaxed for each time of adding electrons. By interpolating the $n_{\text{extra_e}}$ — E_{F} relation, we obtain the $n_{\text{extra_e}}$ for the target E_{F} (converted from the electrode potential). Compared with the CHE model, our approach is closer to the reaction conditions at real electrochemical interfaces.

In response to directly address this comment, we have added explanatory text in our R-MS (Page 21):

'In this work, we use the implicit solvation model as implemented in VASPsol.'

Response to Reviewer #3

Reviewer's Remarks to Authors

This referee summarizes the main achievement of the manuscript as "reported the oxyanion-engineered nickel sites enhancing the urea electrooxidation reaction (UOR) activity via inhibited water oxidation (i.e. oxygen evolution reaction; OER)." This referee thinks the novelty is not sufficient to be publishable in Nature Communications. Their further technical comments are listed as below.

Response

We highly appreciate the Reviewer #3's valuable comments about our manuscript. We would like to emphasize the novelty and significance of our work as follows:

Currently, urea electrooxidation reaction (UOR) still suffer from upper-limit of current density (e.g., most cases were $\sim 140 \text{ mA cm}^{-2}$) and ambiguous mechanisms. In our work, we at first identified the limited current density is because of a) competing oxygen evolution reaction (OER) under a 'large' potential, b) competition adsorption of hydroxyl anion and urea molecule significantly reducing anode electrocatalyst's activity and stability. Following this principle, we developed a series of oxyanion-coordinated nickel catalysts that exhibited nearly 100 % UOR selectivity and a highest UOR current density achieved so far. The novelty of this work include:

1) A record current density for UOR and N-products selectivity on Ni-SO_x powder catalyst. The optimized sulfur oxyanion nickel (Ni-SO_x) powder catalyst exhibited a record current density of $\sim 330 \text{ mA cm}^{-2}$ and a $99.3 \pm 0.4 \%$ N-products selectivity at 1.65 V (vs. RHE). This performance is significantly improved over conventional benchmark nickel-based UOR catalysts accompanying OER, e.g., $\sim 120 \text{ mA cm}^{-2}$ and $82.6 \pm 0.7 \%$ UOR selectivity at the same conditions.

2) Unveiling the origin of ultra-high UOR performance. In situ Synchrotron based X-ray absorption spectroscopy, in situ Attenuated Total Fourier-transform infrared spectroscopy, in situ Raman, and potential-dependent theoretical computations were used to trace and reveal crucial roles of oxyanion dopant for UOR. It was confirmed oxyanion can not only inhibit OH⁻ adsorption and guarantee high coverage of urea reactant on active sites to avoid OER, but also accelerate urea's C-N bond cleavage to form CNO⁻ intermediates for boosting UOR.

3) Proposing a comprehensive mechanism for competitive adsorption behaviour of hydroxyl anion and urea molecule on the catalyst surface toward UOR /OER competition and the dynamic change of Ni active sites during UOR.

In revision, the following explanatory text has been added in R-MS (Page 4, 18):

'This versatile and feasible oxyanion-engineered strategy solves competitive adsorption of organic reactant and hydroxyl anion on the active sites and opens a fresh avenue for the design of high-performance electrocatalysts under large current density operations, and therefore, is expected to be extended to other organic electrooxidation reactions proceeding in aqueous electrolyte to obviate competing OER.'

'We expect that this strategy will aid future research in practical urea electrolysis, and other multi-electron organic molecule oxidation coupled with cathodic hydrogen evolution for overall atomic economy and additional green energy production.'

Comment 3-1

This referee suggest reference electrode calibration must be performed on the same electrolytes to avoid any errors in UOR potential measurements. Ag/AgCl electrodes are oxidized and unstable in alkaline electrolyte and changes potential or current during UOR measurements. Thus this referee advise the author to use the reference electrode Hg/HgO or SCE in alkaline solution.

Response

Fig. R17 CV curves for (a) Ag/AgCl (saturated KCl) electrode and (b) Hg/HgO electrode calibration in 1 M KOH (practical pH=13.50) at 25 °C.

Following the kind suggestion of Reviewer #3, we carried out corresponding electrochemical tests with Hg/HgO as the reference electrode. Prior to testing, Hg/HgO and Ag/AgCl (saturated KCl) reference electrode were calibrated with respect to reversible hydrogen electrode (RHE). The calibration was performed in high purity hydrogen saturated electrolyte with Pt foil as both the working electrode (WE) and counter electrode (CE) at 25 °C. CVs were run at a scan rate of 1 mV s⁻¹, and the average of the two

potentials at which the current crossed zero was taken as the thermodynamic potential for the hydrogen electrode reactions. Three parallel experiments were conducted for each electrode to eliminate accidental error (**Fig. R17 and Table R1**). The calibrated value for Ag/AgCl (saturated KCl) electrode in 1 M KOH (practical pH=13.50) at 25 °C was 0.990 V, which is in good agreement with the value (0.993 V) calculated from the Nernst equation ($E_{RHE} = E_{Ag/AgCl} + 0.059 \times \text{pH} + 0.197$). And the calibrated value for Hg/HgO electrode in 1 M KOH (practical pH=13.50) at 25 °C was 0.890 V, which is in good agreement also with the value (0.894 V) calculated from the Nernst equation ($E_{RHE} = E_{Hg/HgO} + 0.059 \times \text{pH} + 0.098$).

Table R1. Three parallel experiments for calibrating the reference electrode. The potential values are referenced to reversible hydrogen electrode (RHE).

Reference Electrode	Electrolyte	1 st (V)	2 nd (V)	3 rd (V)
Ag/AgCl	1 M KOH	0.992	0.990	0.989
Hg/HgO	1 M KOH	0.892	0.890	0.889

Then, we compare the UOR performance for Ni-SO_x and NiO_x with Hg/HgO and Ag/AgCl (saturated KCl) as reference electrodes, respectively. As is seen in **Fig. R18**, the LSV curves for Ni-SO_x and NiO_x measured with Hg/HgO as the reference electrode are consistent with those of Ag/AgCl (saturated KCl) as the reference electrode. This finding confirms that Ag/AgCl (saturated KCl) electrode is stable in alkaline solution when protected by a salt bridge, that is attributed to the direct contact of Ag/AgCl with the saturated KCl solution rather than the alkaline solution.

Fig. R18 ECSA-normalized LSV curves in 1 M KOH solution containing 0.33 M urea with Ag/AgCl (saturated KCl) electrode and Hg/HgO electrode as reference electrode, respectively.

In response to address directly this comment, **Fig. R18** has been added in R-SI as **Supplementary Fig. 9** with the related discussion on Page 6 and additional text added in R-MS (Page 6, 20):

'As is seen in Supplementary Fig. 9, the LSV curves for Ni-SO_x and NiO_x measured with Hg/HgO as the reference electrode are consistent with those of Ag/AgCl (saturated KCl) as the reference electrode. This finding confirms that Ag/AgCl (saturated KCl) electrode is stable in alkaline solution when protected by a salt bridge, that is attributed to the directly contact of Ag/AgCl with the saturated KCl solution rather than the alkaline solution^{1,2}.

'UOR performance for Ni-SO_x and NiO_x powder electrodes were tested in 1 M KOH with 0.33 M urea solution via a typical three electrode system. Both the Ag/AgCl (saturated KCl) with a salt bridge and Hg/HgO were used as the reference electrodes. As is shown in the electrochemically active surface area (ECSA) normalized linear sweep voltammetry (LSV) curve (Fig. 2a and Supplementary Fig. 8, 9), both Ni-SO_x and NiO_x exhibit higher anodic current toward UOR (solid line) than that for OER (dash line).'

'The Ag/AgCl (saturated KCl) with a salt bridge and Hg/HgO were used as the reference electrodes, while a graphite rod was used as the counter electrode.'

Comment 3-2

To avoid the error in the quantification of UOR/OER selectivity, this referee suggest authors repeat the experiments at least three times and show the data in the error bars (the standard deviation of a data set; Fig. 3c, d).

Response

We agree with Reviewer #3 and, therefore, we performed three independent experiments for the quantification of ion and gas products of Ni-SO_x and NiO_x via ion chromatography (IC) and gas chromatography (GC). As is shown in **Fig. R19**, the data for these three independent experiments have little deviation (error bars), evidencing good reproducibility of the experimental data.

Fig. R19 FEs for different UOR /OER products under different potentials on (a) Ni-SO_x and (b) NiO_x.

In response to address directly this comment, **Fig. R19** has been added in R-MS as **Fig. 3c, d** with the related discussion on Page 9, 10:

'Error bars indicate the standard deviation based on three independent measurements.'

'For Ni-SO_x, the FEs for N-containing products ($FE_{N-products}$) from UOR remained above $95 \pm 4 \%$ at all potentials with few OER products (O_2). In contrast, for NiO_x, the $FE_{N-products}$ monotonically decreased, and FE_{O_2} increased with increasing applied potential. Specifically, UOR selectivity was reduced to $82.6 \pm 0.7 \%$ at 1.65 V for NiO_x, whilst that for Ni-SO_x remained up to $99.3 \pm 0.4 \%$.'

Comment 3-3

The white-line area of NiO is significantly smaller than that of Ni-SO_x (Fig. 4a). Then, the author stated that “with applied potential increasing, the state of Ni increased from +2 and remained below +3 during UOR up to 1.60 V”. This referee recommend authors should measure the white line area and plot the graph between the white line area (similar energy ranges) and the oxidation states at different potentials (see ref. Applied Catalysis B: Environmental, 111–112, 2012, 509-514).

Response

We thank Reviewer #3 for providing this comment. According to the literature (Applied Catalysis B: Environmental, 111–112, 2012, 509-514), the absorption edge at the Rh *K-edge* XANES spectra or white line area intensity at the Rh *L₃-edge* XANES is related to the Rh oxidation state. In our MS, we showed the *in-situ* Ni *K-edge* XANES spectra instead of the Ni *L-edge* XANES in original Fig. 4a. In the case of Ni *K-edge* XANES spectra, it is the position of absorption edge rather than the white line area intensity that manifests the Ni oxidation state (*Nat. Commun.* 2022, 13, 3857, *Nat. Energy* 2021, 6, 904-912, and *Angew. Chem. Int. Ed.* 2019, 58, 1252-1265). Therefore, we measure position of the absorption edge for Ni-SO_x and plot the magnified absorption edge region in the inset of **Fig. R20**. With applied potential increasing, the adsorption edge of Ni-SO_x is positively shifted towards higher energy than that of Ref NiO (+2 valence), but still lower than that of Ref LaNiO₃ (+3 valence). This finding evidence that the state of Ni in Ni-SO_x increase from +2 and remain below +3 during UOR up to 1.60 V. Furthermore, based on the first derivatives of Ni K-edge XANES in Ni-SO_x, Ref NiO, and Ref LaNiO₃, we plot the linear relationship between the Ni K-edge absorption edge and the Ni oxidation states in Ni-SO_x@1.4 V, Ni-SO_x@1.5 V, Ni-SO_x@1.6 V, Ref NiO, and Ref LaNiO₃ (**Fig. R21**). This relationship has been demonstrated in many studies, including, *Sci. Adv.* 2021, 7, eabk0919 and *Nat. Energy* 2021, 6, 1054-1066.

Fig. R20 In situ Ni K-edge XANES spectra for Ni-SO_x under varying potential during UOR. Inset, magnified absorption edge region.

Fig. R21 Relationship between the Ni K-edge absorption edge and the Ni oxidation states in Ni-SO_x under varying potential during UOR, reference NiO and LaNiO₃.

In response to address directly this comment of Reviewer #3, **Fig. R20** has been added in R-MS as **Fig. 4a** and **Fig. R21** has been added in R-SI as **Supplementary Fig. 16**. Also, the following discussion has been added in R-MS (Page 10):

'With applied potential increasing, the adsorption edge of Ni-SO_x is positively shifted towards higher energy than that of Ref NiO, but still lower than that of Ref LaNiO₃, which evidencing that the state of Ni in Ni-SO_x increased from +2 and remained below +3 during UOR up to 1.60 V (Fig. 4a). And a linear relationship between the Ni K-edge absorption edge and the Ni oxidation states in Ni-SO_x@1.4 V, Ni-SO_x@1.5 V, Ni-SO_x@1.6 V, Ref NiO, and Ref LaNiO₃ is plotted (Supplementary Fig. 16)^{30,31}.'

END OF RESPONSE TO REVIEWS

Reviewers' comments:

Reviewer #1 (Remarks to the Author):

The revised version can be accepted.

Reviewer #2 (Remarks to the Author):

Comment 2.4

I still find surprising that N₂ can be generated from urea in oxidizing conditions. Can the authors comment and explain their findings? Without a clear hypothesis, I would rather blame a mistake in the experimental set-up.

Comment 2.5.

I still don't see any reason to substitute a cation by an anion. This won't happen and introduce a fake doping of the system. Those results should be removed.

Providing the distance of S atom from the H atomic layer "during geometric optimization" with a x-axis label 'simulation time' in Fig. R12 once again shows that the authors are performing calculations but lack basic skills in the field.

The negative charge does not seem to be accumulated on the S in Fig R14 but rather in the Ni vacancy. In the SO₄ di-anion, I don't expect the charge to accumulate on the sulfur based on basic electronegativity scale. Again, the authors lack a proper chemical sense.

The material becomes a conductor (see Figure S16). Is this expected? A +U correction is needed for those materials (see 10.1021/acs.jpcc.1c06170).

The sentence "DFT calculations also show that the adsorption strength for OH* on NiOOH-PO₄(O), NiOOH-PO₄(Ni), NiOOH-SeO₄(O), and NiOOH-SeO₄(Ni), are lower than that of NiOOH " disagrees with the data shown in Fig R13.

Response to Reviewer #1

Reviewer's Remarks to Authors

The revised version can be accepted.

Response

We highly appreciate Reviewer #1's positive comments on our manuscript.

Response to Reviewer #2

Comment 2.4

I still find surprising that N₂ can be generated from urea in oxidizing conditions. Can the authors comment and explain their findings? Without a clear hypothesis, I would rather blame a mistake in the experimental set-up.

Response:

We appreciate your valuable feedback and would like to provide a clear explanation for the generation of N₂ from urea in oxidizing conditions.

The valence states of N atoms in urea and N₂ are -3 and 0 respectively, so it is easy to oxidize urea to N₂ under oxidation conditions. And we have indeed detected the real-time generated N₂ from urea oxidation in the experiment. The generated N₂ may follow the equation: $\text{CO}(\text{NH}_2)_2 + \text{OH}^- \rightarrow \text{CO}_2 + \text{N}_2 + \text{H}_2\text{O} + 6\text{e}^-$, and the typical reaction pathway: urea adsorption; dehydrogenation of N-H; C-N bond breakage; N-N coupling; N₂ and CO₂ desorption. This phenomenon of urea oxidation to N₂ in oxidizing conditions and the typical reaction pathway has also been reported and validated in many studies *via* experiments or DFT calculations including *ACS Catal.*, 2022, 12, 569-579, *Angew. Chem. Int. Ed.* 2022, 61, e202209839, *Angew. Chem. Int. Ed.* 2021, 60, 7297-7307, and *J. Phys. Chem. A*, 2010, 114, 11513-11521.

As for the experimental set-up, we have also referred to some studies (*Angew. Chem. Int. Ed.* 2022, 61, e202209839, *Nat. Energy*, 2021, 6, 904-912) and set a rigorous protocol, we have:

- 1) analyzed N₂ signal using on-line gas chromatography (GC) every 15 min;
- 2) repeated N₂ measurements during each applied potential at least 3 times to exclude any possible errors. Error bars have been added in first round revised manuscript (Fig. 3c and d);
- 3) real-time detected N₂ during electrochemical curve test (CV) utilizing *in situ* differential electrochemical mass spectrometry (DEMS) (first round revised manuscript, Fig. 3a and b). These on-line/real-time N₂ detection methods and multiple repeated measurements ensure the accuracy and reliability of our experimental set-up.

In response to directly address this comment, we have added a clear explanation for the generation of N₂ from urea in oxidizing conditions and the details of experimental set-up in our revised

manuscript (R-MS, Page 9, 18-20):

'The generated N₂ may follow the equation: CO(NH₂)₂ + OH⁻ → CO₂ + N₂ + H₂O + 6e⁻, and the typical reaction pathway: urea adsorption; dehydrogenation of N-H; C-N bond breakage; N-N coupling; N₂ and CO₂ desorption^{10,12,16,21}.'

'Real-time generated gaseous products (N₂ and O₂) during the Cyclic voltammetry (CV) test were pumped to in situ DEMS system (HPR-40, Hiden Analytical Ltd.), and the signal with mass-to-charge ratio of 28, 32, was obtained'

'The gas products (N₂, O₂) formed during urea oxidation were analyzed every 15 minutes via on-line gas chromatograph (GC, 8890, Agilent). Measurements for all products were repeated at least 3 times during each applied potential to exclude any possible errors.'

Comment 2.5

I still don't see any reason to substitute a cation by an anion. This won't happen and introduce a fake doping of the system. Those results should be removed.

Response

Following suggestion, we have removed calculations that substituted a cation by an anion. Please refer to the revised Fig. 6.

Providing the distance of S atom from the H atomic layer "during geometric optimization" with a x-axis label 'simulation time' in Fig. R12 once again shows that the authors are performing calculations but lack basic skills in the field.

Response:

We are sorry about the misunderstanding arising from the data in Fig. R12. The intention behind Fig. R12 was to provide a comprehensive visualization of the dynamic optimization process of the S substitution O structure. Following the Reviewer's feedback, we now recognize that this approach to presenting the data, the distance of S atom from the H atomic layer with 'simulation time' as the x-axis label in Fig. R12b, might have led to confusion.

Therefore, we provide the energy change from the initial to the final ionic step for the S substitution O structure during structure optimization, as is seen in **Fig. R1**, which we believe offers a more conventional and clear understanding of the geometric optimization process.

Furthermore, as Fig. R12b in first round revised manuscript does not directly contribute to the construction of the model, we have removed it to eliminate possible misunderstandings by the reviewer or readers.

Fig. R1 The energy versus structure optimization step for the S substitution O structure during structure optimization. The snapshots in the figure show the structure of the initial and final steps.

The negative charge does not seem to be accumulated on the S in Fig. R14 but rather in the Ni vacancy. In the SO_4 di-anion, I don't expect the charge to accumulate on the sulfur based on basic electronegativity scale. Again, the authors lack a proper chemical sense.

Response:

We respectfully disagree with this comment. This seems to be a misunderstanding caused by the Reviewer's misinterpreted " SO_4 group" as "S" atom. In the first-round revision (manuscript, page 13, lines 20-22), we described the charge accumulation on "sulfur oxyanion groups (SO_4)" which is distinct from the individual "S" atom itself. " SO_4 group" was considered as a whole in our calculations because S had been oxidized to SO_4 groups on the surface of the Ni- SO_x catalyst (R-MS, Fig. 1). More importantly, our purpose of constructing NiOOH- SO_4 model is to verify the effect of SO_4 -doping instead of S-doping on the adsorption of OH^- in nickel active sites.

Fig. R2 Differential charge density for NiOOH- SO_4 (a) top view (b) side view. The yellow and cyan colors represent the charge accumulation and depletion regions, respectively (isovalue, 0.005). (c) charge accumulation and depletion of each atom on the $-\text{SO}_4$ group obtained by Bader charge analysis.

In addition, to give the Reviewer a better understanding of charge accumulation/depletion on the " SO_4 group" itself, we have provided a detailed Bader charge analysis. As is seen in Fig. R2, charge

depletion on “S” atom of "SO₄ group", and charge accumulation on “O” atom of "SO₄ group", but the charge accumulation on overall “SO₄ group” is 1.537e⁻. This result is in good agreement with our conclusion regarding “the charge accumulation on sulfur oxyanion groups”, and is consistent with electronegativity scale.

In response to fully address this comment, **Fig. R2** has been updated as **Supplementary Fig. 19** in the latest revised supporting information (**R-SI**) with the related discussion on Page 11.

‘As is seen in Supplementary Fig. 19c, charge depletion on “S” atom of "SO₄ group", and charge accumulation on “O” atom of "SO₄ group", but the charge accumulation on overall “SO₄ group” is 1.537e⁻.’

The material becomes a conductor (see Figure S16). Is this expected? A +U correction is needed for those materials (see 10.1021/acs.jpcc.1c06170).

Response:

We agree that the material is a conductor – an observation consistent with our electrocatalytic experimental (Fig. 2 in R-MS) and calculated results (Fig. 6 in R-MS).

Here we want to emphasize that all our calculations have been performed with +U correction in the previous revision. Following the suggestion, we have further amended our latest revised manuscript to clearly indicate this point of view. Specifically, we have used a U_{eff} value of 5.5 eV, which aligns with the reference you kindly provided (10.1021/acs.jpcc.1c06170).

In response to fully address this comment of the reviewer, we have added the following description in the R-MS (page 20).

‘All structures in the calculations were explored using PBE+U, with an effective $U_{\text{eff}} = U - J$ term of 5.5 eV, for the Ni 3d state⁴⁷⁻⁴⁹,’

The sentence "DFT calculations also show that the adsorption strength for OH* on NiOOH-PO₄(O), NiOOH-PO₄(Ni), NiOOH-SeO₄(O), and NiOOH-SeO₄(Ni), are lower than that of NiOOH " disagrees with the data shown in Fig R13.

Response:

We respectfully disagree with this comment. In these calculations, higher OH* adsorption energy values (ΔE_{OH^*}) indicate weaker OH* adsorption strength. The ΔE_{OH^*} of NiOOH-PO₄(O), NiOOH-PO₄(Ni), NiOOH-SeO₄(O), and NiOOH-SeO₄(Ni) were significantly higher than that of NiOOH. Consequently, OH* adsorption strength on these materials are lower (weaker) than that of NiOOH, which is in good agreement with the data shown in Fig. R13.

End of response

REVIEWER COMMENTS

Reviewer #4 (Remarks to the Author):

The manuscript reports a very active urea oxidation catalyst derived from the in-situ oxidation of NiS₂. It would also be of interest to know which role the urea plays for the formation of the actual catalyst: Would the resulting material be very different if produced in pure KOH? - And what about Ni oxidised in the presence of sulfate ions, would this lead to a similar catalyst as the one derived from NiS₂?

As far as I can tell, the experimental work looks proper and credible and the proposed material could interest a reasonably broad audience, both in terms of performance, but also in terms of catalyst preparation method.

The authors also provide a small computational study that is supposed to support the difference between two materials "standard" NiOOH and NiOOH that is derived from the oxidation of NiS₂. The problem, is that (a) already the geometry of "standard" NiOOH is debated in the literature, with the model that the authors adopted (all protons on the "inner" side, i.e., not exposed to the solvent) being surprising. (b) the choice for NiOOH-SO₄ is more than surprising: simply adding SO₄ (or formally SO₃) to the surface model is, in my opinion, completely irrelevant, in the sense that there are no indications that the experimentally produced material looks anything close to this one. From the HRTEM, one has the impression that the oxidised surface layer of NiS₂ is essentially amorphous. So yes, locally (as probed by Raman) it might resemble NiOOH, but to model this surface by a perfect crystal plane with one protruding SO₄ group is quite far fetched.

The second significant issue with the computational work is that the authors focus on the OH adsorption energy (which, by the way, could have been computed with the more common GC-DFT method, instead of the approximate charge-based method). For a catalytic reaction (not) to occur, the full reaction pathway using a relevant surface state should be investigated, not only one step. Hence, in the current state, it is not possible to derive a predicted thermodynamic onset potential. Furthermore, the NiOOH OER mechanism itself is heavily debated. Hence, it will be difficult to be convincing on this aspect. Even the steps that are discussed in Fig. 7 c and d are not computed, i.e., the (local) formation of Ni(IV)O(OH)₂ or the Ni(II)(OH)₂.

In summary, the computational evidence provided does not significantly support the mechanistic proposal for Fig. 7 and is, with its somewhat odd structural models, not convincing.

Response to Reviewer #4

Comment 4.1

The manuscript reports a very active urea oxidation catalyst derived from the in-situ oxidation of NiS_2 . It would also be of interest to know which role the urea plays for the formation of the actual catalyst: Would the resulting material be very different if produced in pure KOH? - And what about Ni oxidised in the presence of sulfate ions, would this lead to a similar catalyst as the one derived from NiS_2 ? As far as I can tell, the experimental work looks proper and credible and the proposed material could interest a reasonably broad audience, both in terms of performance, but also in terms of catalyst preparation method.

Response:

We appreciate Reviewer's positive comments on our manuscript. Urea plays a role in controlling the surface activation process of the catalyst and avoids severe and uncontrollable surface reconstruction of Ni-SO_x . As is shown in Fig. 1b in the original manuscript, when NiS_2 was activated in 1 M KOH solution with 0.33 M urea, it exhibited a regular core-shell structure with a crystalline NiS_2 core and a thick amorphous shell. But when NiS_2 was activated in pure 1 M KOH solution, it lost this regular core-shell structure and suffered severe and uncontrollable surface reconstruction to massive amorphous morphology (Fig. R1a and b). Such severe surface reconstruction would inhibit the adsorption of small organic molecules such as urea, methanol, etc., and lead to side reaction, OER. In addition, when Ni(OH)_2 catalyst was oxidized in the electrolyte containing 0.05 M sulfate ions, the morphology of obtained catalyst was similar to that of NiO_x rather than Ni-SO_x (Fig. R1c and d).

Fig. R1 (a) and (b) Morphology of NiS_2 activated in pure 1 M KOH solution. (c) and (d) Morphology of Ni(OH)_2 catalyst activated in the electrolyte containing 0.05 M sulfate ions.

Comment 4.2

The authors also provide a small computational study that is supposed to support the difference between two materials "standard" NiOOH and NiOOH that is derived from the oxidation of NiS₂. The problem, is that (a) already the geometry of "standard" NiOOH is debated in the literature, with the model that the authors adopted (all protons on the "inner" side, i.e., not exposed to the solvent) being surprising. (b) the choice for NiOOH-SO₄ is more than surprising: simply adding SO₄ (or formally SO₃) to the surface model is, in my opinion, completely irrelevant, in the sense that there are no indications that the experimentally produced material looks anything close to this one. From the HRTEM, one has the impression that the oxidised surface layer of NiS₂ is essentially amorphous. So yes, locally (as probed by Raman) it might resemble NiOOH, but to model this surface by a perfect crystal plane with one protruding SO₄ group is quite far fetched.

Response

We agree with the Reviewer that geometry of "standard" NiOOH is debated in the literature. There are many different variants of the NiOOH structure. Due to the limitation of computational cost and efficiency, we can only screen out the likely and cost-effective NiOOH model according to the experimental data. Although the inability to observe crystal plane of amorphous NiOOH by HRTEM has limited the construction of models, *in situ* extended X-ray absorption fine structure (EXAFS) measurements for catalysts provide a possible solution. Many studies (e.g. *Nat. Commun.* 2022, 13, 2916; *Adv. Funct. Mater.* 2021, 31, 2100614) have reported that *in situ* EXAFS measurements can be used to assisted-guide the DFT model construction of amorphous NiOOH. Therefore, we constructed a relatively matching NiOOH model based on *in situ* EXAFS data. The bulk phase structure of β -NiOOH is referenced to the structure type of EE (hydrogen atoms distributed on both sides) reported by Kowalski et al. and Carter et al (*Nat. Commun.* 2023, 14, 3498 and *Chem. Mater.* 2018, 30, 5205-5219). And as shown in **Fig. R2a**, the simulated spectrum of the backscattering signal χ^3 (blue line) of the β -NiOOH (111) model in the inset matches well with the EXAFS data (grey line) of NiO_x, evidencing the relative rationality of this NiOOH model. Similarly, the simulated spectrum of the backscattering signal χ^3 (red line) of the β -NiOOH (111)-SO₄ model also matches well with the EXAFS data (grey line) of Ni-SO_x, which also confirms the relative rationality of this NiOOH-SO₄ model (**Fig. R2b**).

In response to fully address this comment, **Fig. R2** has been added in the latest revised-supplementary information (R-SI) as **Supplementary Fig. 17** with the related discussion on Page 10: *'Although the inability to observe crystal plane of amorphous NiOOH by HRTEM has limited the construction of models, in situ extended X-ray absorption fine structure (EXAFS) measurements for catalysts provide a possible solution to assisted-guide the DFT model construction of amorphous*

NiOOH^{3-4} . Therefore, we construct a relatively matching NiOOH model based on in situ EXAFS data. The bulk phase structure of $\beta\text{-NiOOH}$ is referenced to the structure type of EE (hydrogen atoms distributed on both sides)^{5,6}. And as shown in **Supplementary Fig. 17a**, the simulated spectrum of the backscattering signal χ^3 (blue line) of the $\beta\text{-NiOOH}$ (111) model in the inset matches well with the EXAFS data (grey line) of NiO_x , evidencing the relative rationality of this NiOOH model. Similarly, the simulated spectrum of the backscattering signal χ^3 (red line) of the $\beta\text{-NiOOH}$ (111)- SO_4 model also matches well with the EXAFS data (grey line) of Ni-SO_x , which also confirms the relative rationality of this NiOOH-SO_4 model (**Supplementary Fig. 17b**).

Fig. R2 (a) EXAFS fitting results of Ni K-edge at k-space of NiO_x during UOR under 1.5 V (vs. RHE). The inset shows the relatively matching NiOOH model. (b) EXAFS fitting results of Ni K-edge at k-space of Ni-SO_x during UOR under 1.4 V (vs. RHE). The inset shows the relatively matching NiOOH-SO_4 model.

Comment 4.3

The second significant issue with the computational work is that the authors focus on the OH adsorption energy (which, by the way, could have been computed with the more common GC-DFT method, instead of the approximate charge-based method). For a catalytic reaction (not) to occur, the full reaction pathway using a relevant surface state should be investigated, not only one step. Hence, in the current state, it is not possible to derive a predicted thermodynamic onset potential.

Response

We are sorry about the misunderstanding arising from the application of the computational method. The method we used is one of the common GC-DFT methods that have been reported in many studies (*Chem. Rev.* 2022, 122, 10675-10709; *J. Am. Chem. Soc.* 2021, 143, 9423-9428; *J. Am. Chem. Soc.* 2022, 144, 17140-17148). We fix the Fermi level of the system as a constant set by the electrode

potential. During the structure optimization process, the number of electrons matching the applied potential is self-consistently solved simultaneously to reach the target potential while obtaining an equilibrium structure.

We agree with the Reviewer that the full reaction pathway should be calculated, investigating whether a catalytic reaction occurs. Based on the NiOOH and NiOOH-SO₄ models shown in Fig. R2, we calculated the possible OER complete pathways. As shown in Fig. R3, the potential-determination step is *OH + OH⁻ → *O + H₂O + e⁻. NiOOH-SO₄ exhibits a larger Gibbs energy barrier of 0.88 eV in comparison to 0.77 eV for NiOOH at 1.5 V applied potential, indicating that OER is more difficult to occur on NiOOH-SO₄. This result is in good agreement with the experimental results without O₂ generation on Ni-SO_x (Original Fig. 3). It should be noted that, as the Reviewer mentioned in comment 4.4, the OER mechanism of NiOOH is debated. Due to the limitation of calculation cost and time, we currently only consider calculating possible OER pathways of NiOOH materials based on the widely reported adsorption evolution mechanism (AEM).

Fig. R3 Gibbs energy profiles of OER in the alkaline (pH = 14) on NiOOH and NiOOH-SO₄ catalysts at U = 1.5 V vs RHE.

Furthermore, based on the observed results of C–N bond cleavage and CNO⁻ intermediate observed by *in situ* ATR-IR, we speculated possible pathways of urea oxidation to main product NO₂⁻. As shown in Fig. R4, the Gibbs energy barrier for C–N bond cleavage on NiOOH-SO₄ is much smaller than NiOOH, which demonstrates favourable C–N bond cleavage on NiOOH-SO₄. This finding is consistent with *in situ* ATR-IR data (Original Fig. 5a and b).

Fig. R4 Gibbs energy profiles of UOR in the alkaline (pH = 14) on NiOOH and NiOOH-SO₄ catalysts at U = 1.5 V vs RHE. The structures of the key intermediates in the reaction process are shown in Supplementary Table 3 in R-SI.

In response to fully address this comment, **Fig. R4** has been added in R-SI as **Supplementary Fig. 18** with the related discussion in the latest revised-manuscript (R-MS) (Page 13).

'Then, we constructed relatively matched NiOOH-SO₄ and NiOOH models to simulate active sites of Ni-SO_x and NiO_x during UOR and speculated possible pathways of urea oxidation to main product NO₂⁻ based on the observed intermediate (Supplementary Fig. 17). The computational results show that C–N bond cleavage is promoted on NiOOH-SO₄ compared with NiOOH, which is consistent with in situ ATR-IR data (Supplementary Fig. 18).'

Some computational limitations have been discussed in Supplementary information section. The DFT computation method and models can be found in Supplementary Table 3 and Supplementary note 1 (Pages 21-31 in R-SI).

Comment 4.4

Furthermore, the NiOOH OER mechanism itself is heavily debated. Hence, it will be difficult to be convincing on this aspect. Even the steps that are discussed in Fig. 7 c and d are not computed, i.e., the (local) formation of Ni(IV)O(OH)₂ or the Ni(II)(OH)₂.

Response

We agree with the Reviewer that the NiOOH OER mechanism is debated. Based on the Reviewer' and Editor's suggestions, we have modified the OER process schematic (**Fig. R5**). We removed the possible dynamic evolution process of NiOOH during OER (NiOOH to NiO(OH)₂ or NiOO₂ and possible pathways for OER (**Fig. R3**), only kept the first step for OER, the adsorption of OH⁻, which

is generally recognized in any OER mechanism under alkaline conditions and in good agreement with our *in situ* ATR-IR spectroscopy results (Original Fig. 5c and d).

Fig. R5 (a) Representation for UOR process on Ni-SO_x accompanying adsorption of reactants and dynamic evolution of active sites under high potential (> 1.50 V). (b) Representation of UOR and OER process on NiO_x accompanying adsorption of reactants and dynamic evolution of active sites under high potential (> 1.50 V).

In response to fully address this comment, **Fig. R5** has been updated as **Fig. 6c and d** in R-MS with the related discussion on Page 14.

‘Based on above electrochemical tests and in situ spectroscopic studies, a possible UOR mechanism on two kinds of electrocatalysts including adsorption of reactants and dynamic evolution of active sites is proposed.’

‘Once the potential is applied over 1.50 V (Fig. 7c, d), derived NiOOH is attacked by OH⁻ to generate high valance Ni species (≥ 3).’

End of Response